# Generating Multi-Table Time Series EHR from Latent Space with Minimal Preprocessing

**Eunbyeol Cho**[1], **Jiyoun Kim**[1], **Minjae Lee**[2], **Sungjin Park**[1], **Edward Choi**[1]

KAIST[1]   FuriosaAI[2]

{eunbyeol.cho,edwardchoi}@kaist.ac.kr[1]

## Abstract

Electronic Health Records (EHR) are time-series relational databases that record patient interactions and medical events over time, serving as a critical resource for healthcare research and applications. However, privacy concerns and regulatory restrictions limit the sharing and utilization of such sensitive data, necessitating the generation of synthetic EHR datasets. Unlike previous EHR synthesis methods—which typically generate medical records consisting of expert-chosen features (*e.g.*, a few vital signs, structured codes only)—we introduce RawMed, the first framework to synthesize multi-table, time-series EHR data that closely resembles raw EHRs. Using text-based representation and compression techniques, RawMed captures complex structures and temporal dynamics with minimal lossy preprocessing. We also propose a new evaluation framework for multi-table time-series synthetic EHRs, assessing distributional similarity, inter-table relationships, temporal dynamics, and privacy. Validated on two open-source EHR datasets, RawMed outperforms baseline models in fidelity and utility. The code is available at https://github.com/eunbyeol-cho/RawMed.

## 1 Introduction

The digitalization of medical data has accelerated the adoption of Electronic Health Records (EHR), one of the most significant innovations in modern healthcare. EHRs systematically store various medical events, such as prescriptions and test results, along with corresponding timestamps in a multi-table relational database, encompassing categorical, numerical, and text data. Due to these properties, EHRs serve as a crucial resource in various medical AI studies, including clinical prediction, information retrieval, and question answering [1, 2].

However, since EHRs contain sensitive personal information, they are subject to privacy regulations that hinder data sharing and research applications [3]. In response, synthetic data has emerged as a promising solution, offering a privacy-preserving alternative that can still support research and clinical applications [4, 5]. Progress in generative models, especially Generative Adversarial Networks (GANs) [6], has opened up new avenues for synthesizing EHR data. Early efforts primarily generated discrete features or heterogeneous features, often overlooking the time-series aspect [7, 8]. More recent studies incorporate both time-series and heterogeneous features, gradually approaching a closer reflection of real EHR data [9, 10, 11, 12].

Despite this progress, existing EHR synthesis studies still face the following limitations. **First, there is a high dependence on feature selection**. Most approaches [9, 10, 12] typically select only a subset of tables and columns of the entire EHR database based on domain knowledge, and generate synthetic data exclusively for the features within those selected columns.[1] While this method

---

[1]In this paper, a *column* denotes a raw data field in an EHR table, while a *feature* refers to a processed data representation derived from one or more columns. See Appendix A.1.

39th Conference on Neural Information Processing Systems (NeurIPS 2025).

simplifies the synthesis process, it limits the usefulness of the generated data. For example, if a new research question or analysis requires a feature that was excluded, it is impossible to build predictive models or perform statistical analyses based on that synthetic EHR data. Morever, prior research [13, 14, 15] suggests that models incorporating a broader range of features tend to achieve higher predictive accuracy, implying that richer synthetic data offers wider utility and more substantial value across various downstream tasks. **Second, many existing approaches rely on complex, lossy preprocessing steps.** Techniques such as numeric binning, term normalization, and aggregation are commonly used, but can unintentionally distort the data or obscure meaningful patterns. For instance, aggregating lab values over time may mask sudden anomalies while binning them into discrete ranges may oversimplify subtle trends, reducing the synthetic data's fidelity for predictive modeling.

To address these challenges, our framework, `RawMed`, synthesizes raw EHRs, multi-table time-series data that preserve all columns and original values in their database form, as illustrated in Figure 1. To implement this approach, `RawMed` adopts a text-based method, treating EHR data as text to retain original values without additional transformations (e.g., binning, aggregation, or term normalization). This not only reduces the risk of data distortion, but also makes our method generalizable to various EHR data with different database schemas. However, as time-series data becomes longer, its textual representation typically grows even longer due to subword tokenization, leading to a significant increase in computational complexity.

Table 1: Maximum number of features and time steps across datasets for each method. [1]: Number of features (see Appendix A), [2]: Number of time steps, [3]: Uses all columns, [4]: Preserves original values.

| Method | [1] | [2] | [3] | [4] |
|---|---|---|---|---|
| [9] | 98 | 24 | × | × |
| [10] | 90 | 50 | × | × |
| [11] | 5,373 | 48 | × | × |
| [12] | 15 | 276 | × | × |
| `RawMed` | 333,524 | 243 | √ | √ |

To mitigate this, we employ Residual Quantization [16] to compress textualized time-series data, enabling autoregressive modeling in a compressed latent space. This reduces sequence length and computational complexity, allowing `RawMed` to handle multi-table EHR datasets with numerous columns. Consequently, our framework generates raw EHRs with extensive features and time steps surpassing most existing methods (Table 1). Our key contributions are summarized as follows:

1. **First multi-table and time-Series EHR generation.** This study introduces `RawMed`, the first framework for generating raw multi-table time-series EHRs, preserving all columns and original values, and demonstrating the method's feasibility with three primary tables.
2. **Novel evaluation framework for synthetic raw EHRs.** Unlike prior work that synthesizes only a subset of features, `RawMed` generates all elements of raw EHRs, making the quality assessment more challenging. We propose a comprehensive evaluation framework encompassing low-order statistics, downstream utility, multi-table interactions, time-series fidelity, and privacy protection.
3. **Open-source validation and code release.** We validated `RawMed` on two open-source EHR datasets, confirming its effectiveness, and plan to release all source code to support further research in synthetic EHR generation.

## 2 Related work

### 2.1 EHR data generation

Early research on EHR synthesis primarily focused on generating single data types (*e.g.*, diagnostic codes) using generative models such as GANs and VAEs. Subsequent studies explored mixed data types or incorporated time-series continuity [17, 18, 19, 20, 21, 22]. However, relatively few studies have addressed both the temporal dynamics and heterogeneous nature of EHR data simultaneously, a gap that has only recently started to gain attention. For example, EMR-M-GAN [9] pioneered a GAN-based approach for mixed-type time-series EHR data, using a Dual-VAE and Coupled Recurrent Network (CRN) to capture inter-feature correlations and temporal dynamics. EHR-Safe [10] also adopted a GAN-based framework designed to handle real-world EHR complexities comprehensively, including missing data, diverse feature types, and both static and time-varying attributes. FLEXGEN-EHR [11] integrated static and time-series data, focusing on missing values via optimal transport. TIMEDIFF [12] introduced a hybrid diffusion model capable of generating both continuous and discrete time-series data concurrently. To the best of our knowledge, no existing studies utilize all

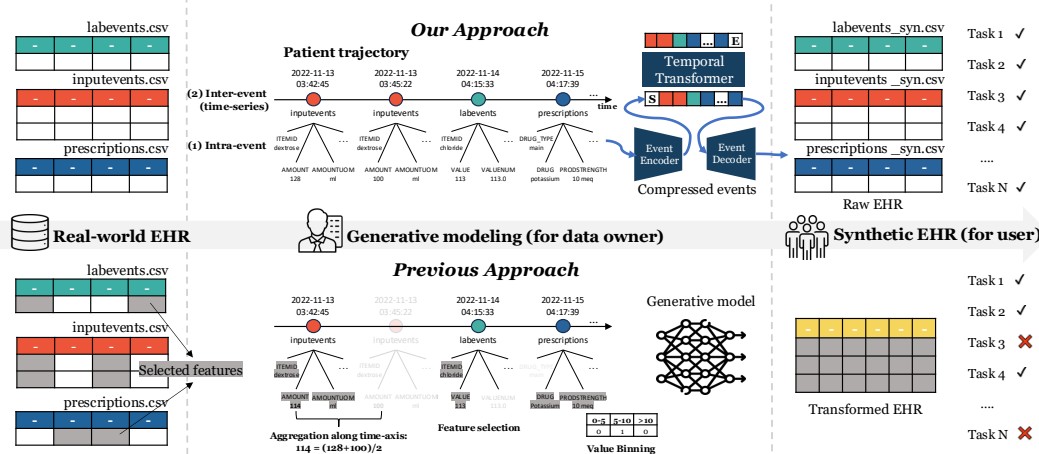

Figure 1: Conceptual overview of the `RawMed` pipeline. **Left**: Real-world EHR data. **Center**: Data generation process. **Right**: Resulting synthetic data. The **bottom** illustrates conventional approach focused on feature selection and domain-specific engineering, while the **top** illustrates `RawMed`, which minimizes domain-dependent preprocessing to generate raw EHR-like data, thereby enhancing user flexibility and utility.

columns from the original EHR database tables. Instead, they rely on a subset of features deemed important by the researchers, and heavily preprocess (*e.g.*, numerical binning, aggregation) the original EHR data to train the generative model.

## 2.2 Text-based approaches for tabular data generation

With the rapid progress of NLP, studies have emerged that convert tabular data into textual form for generation purposes [23, 24]. This text-based approach offers several advantages: it minimizes complex preprocessing, preserves original values without information loss, and leverages the pre-trained knowledge of language models. Such benefits are particularly evident for datasets like EHRs, which often follow varying database schemas and require substantial preprocessing. Nevertheless, most existing research on tabular data synthesis focuses on single-table settings [23, 24], with comparatively few studies addressing relational database [25] and no studies synthesizing time-series relational database. Moreover, converting tabular data into text increases the length of each row due to subword tokenization, potentially inflating training and inference costs. To mitigate this, `RawMed` builds on text-based representations but compresses textualized data, enabling efficient autoregressive modeling in a latent space. This approach reduces computational cost while preserving the fidelity of raw, multi-table time-series EHRs across diverse database schemas.

## 3 Method

`RawMed` is a framework for synthesizing multi-table time-series EHRs, capturing complex patient trajectories with minimal lossy preprocessing. Its architecture supports heterogeneous data type and varying event counts using text-based event representation, with two modules: (1) event-level compression for compact event encoding and (2) temporal inter-event modeling for dynamic temporal relationships. This design enables precise synthesis of large-scale raw EHR datasets.

### 3.1 Data representation and notation

When admitted to a hospital, patients typically undergo a series of clinical events, such as laboratory tests and medication administrations, which are recorded as individual rows in relational tables (*e.g.*, *lab*, *medication*). Each table includes columns such as *timestamp* indicating when the event occurred, *item* identifying a specific clinical measurement (*e.g.*, glucose) or a drug name (*e.g.*, propofol), and *value* (*e.g.*, 95) and *uom* (unit of measure, *e.g.*, mg/dL) for event details. By ordering these events chronologically, a patient's clinical trajectory is represented as a time-ordered sequence.

Formally, for a patient $p \in \mathcal{P}$, the event sequence is $S^p = [e_1^p, e_2^p, \ldots, e_{n^p}^p]$, where $n^p$ is the number of events. Each event $e_i^p$ consists of a timestamp $t_i^p$, representing time elapsed since hospital admission, an event type $\epsilon_i^p$ (e.g., *lab*), and a set of column-value pairs $a_i^p = \{(c_j, v_j) \mid j = 1, \ldots, m^{\epsilon_i^p}\}$, where $c_j$ is the column name, $v_j$ its textualized value, and $m^{\epsilon_i^p}$ the number of columns for event type $\epsilon_i^p$. Although the *timestamp* is a column in the relational table, we isolate $t_i^p$ from $a_i^p$ to highlight its role in temporal ordering.

To create a textual representation, we serialize each event by concatenating the table name $\epsilon_i^p$, followed by column names and their values, excluding null-valued columns. For example, an event from the *lab* table with $a_i^p = \{(\text{item}, \text{"Glucose"}), (\text{value}, \text{"95"}), (\text{uom}, \text{"mg/dL"})\}$ is serialized as "lab item Glucose value 95 uom mg/dL". Thus, an event is denoted as $e_i^p = (t_i^p, x_i^p)$, where $x_i^p$ is the serialized text string. Then $x_i^p$ is tokenized, padded or truncated to a fixed length $L$ (typically 128), and embedded to $\mathbf{x}_i^p \in \mathbb{R}^{L \times F}$. Given the high dimensionality of these embeddings, compression is essential to model the entire patient sequence $S^p$, a challenge also addressed in the context of text-based EHR embeddings [26].

## 3.2 Compressing event representations

To address this, `RawMed` compress the event text embedding $\mathbf{x}_i^p \in \mathbb{R}^{L \times F}$, where $L$ is the number of tokens and $F$ is the embedding dimension, into a discrete latent representation $\mathbf{z}_i^p \in \mathbb{R}^{L_z \times F_z}$, where $L_z$ is the latent sequence length and $F_z$ is the latent embedding dimension. We adopt neural network-based compression methods, Vector Quantized Variational AutoEncoders (VQ-VAE) [27] and Residual Quantization (RQ) [16], implemented with 1D convolutional neural networks (CNNs).

**Encoder and quantization** The encoder `Enc` transforms $\mathbf{x}_i^p$ into $\hat{\mathbf{z}}_i^p = \text{Enc}(\mathbf{x}_i^p) \in \mathbb{R}^{L_z \times F_z}$, consisting of $L_z$ latent vectors. Each latent vector, denoted $\hat{\mathbf{z}} \in \mathbb{R}^{F_z}$ for simplicity, is quantized to $\mathbf{z} \in \mathbb{R}^{F_z}$ by assigning it to the nearest entry in the codebook $C = \{(k, \text{lut}(k)) \mid k = 1, \ldots, K\}$, where $\text{lut}(k) \in \mathbb{R}^{F_z}$ is the embedding for index $k$.[2] The quantization is defined as:

$$\text{VQ}(\hat{\mathbf{z}}; C) = \underset{k \in [K]}{\arg\min} \|\hat{\mathbf{z}} - \text{lut}(k)\|_2^2, \quad \mathbf{z} = \text{lut}\big(\text{VQ}(\hat{\mathbf{z}}; C)\big).$$

Thus, the quantized latent representation $\mathbf{z}_i^p \in \mathbb{R}^{L_z \times F_z}$ comprises $L_z$ number of $\mathbf{z}$ vectors. To enhance the expressiveness of the representation, Residual Quantization (RQ) decomposes each latent vector $\hat{\mathbf{z}} \in \mathbb{R}^{F_z}$ from $\hat{\mathbf{z}}_i^p$ into multiple quantized components. Specifically, RQ represents $\hat{\mathbf{z}}$ as a tuple of indices and composes it as:

$$\text{RQ}(\hat{\mathbf{z}}; C, D) = (k_1, \ldots, k_D) \in [K]^D, \quad \mathbf{z} = \sum_{d=1}^{D} \text{lut}(k_d),$$

where $D$ is the quantization depth (*i.e.*, $D{=}1$ for VQ). The RQ process initializes the residual vector as $\mathbf{r}_0 = \hat{\mathbf{z}}$. For each depth $d = 1, \ldots, D$, it quantizes the residual to obtain an index $k_d = \text{VQ}(\mathbf{r}_{d-1}; C)$ and computes the next residual as $\mathbf{r}_d = \mathbf{r}_{d-1} - \text{lut}(k_d)$. Partial sums $\mathbf{z}^{(d)} = \sum_{m=1}^{d} \text{lut}(k_m)$ yield increasingly refined approximations, with $\mathbf{z} = \mathbf{z}^{(D)}$.

**Decoder and loss function** The decoder `Dec` reconstructs the original embedding as $\hat{\mathbf{x}}_i^p = \text{Dec}(\mathbf{z}_i^p)$. The model is trained by minimizing a combination of the reconstruction loss $\|\mathbf{x}_i^p - \hat{\mathbf{x}}_i^p\|_2^2$ and the commitment loss. The loss of commitment encourages $\text{Enc}(\mathbf{x})$ to remain close to $\mathbf{z}$ used during quantization. Details on architecture and training are provided in Appendices D.1 and E.

## 3.3 Temporal modeling between events

To model temporal relationships between events $e_i^p$, we transform the patient's trajectory $S^p$ by replacing each event's text embedding $\mathbf{x}_i^p$ with its compressed representation $\mathbf{z}_i^p$. Each $\mathbf{z}_i^p$ is mapped to a sequence of $L_z \cdot D$ discrete indices $k_i^p \in [K]^{L_z \times D}$, forming the compressed trajectory $S_{\text{quantized}}^p = [(t_i^p, k_i^p) \mid i = 1, \ldots, n^p]$, where $t_i^p$ is the timestamp of the $i$-th event.

---

[2]The function `lut` refers to a look-up table that maps indices to their corresponding embeddings.

**Time tokenization**  Each timestamp $t_i^p$ is tokenized into a fixed-length sequence of $L_t$ digits, denoted $\tau_i^p \in \{0, 1, \ldots, 9\}^{L_t}$. With a 10-minute resolution, $t_i^p$ is divided by 10 and decomposed into its tens, hundreds, and (if needed) thousands digits (*e.g.*, 720 minutes becomes $[7, 2]$). This provides explicit temporal context alongside event content.

**Sequence representation and autoregressive modeling**  Each event $(\tau_i^p, k_i^p)$ is represented as a block of tokens, combining the $L_t$ time tokens from $\tau_i^p$ with the $L_z \cdot D$ flattened indices from $k_i^p$. Concatenating these blocks yields a unified sequence:

$$S_{\text{quantized}}^p = [\tau_1^p, k_1^p, \tau_2^p, k_2^p, \ldots, \tau_{n^p}^p, k_{n^p}^p],$$

with a total length of $n^p \times (L_t + L_z \cdot D)$. This sequence integrates temporal and event-based information, enabling modeling temporal dependencies effectively. We use a Transformer-based model, `TempoTransformer`, to autoregressively predict each token based on its predecessors. The training objective minimizes the negative log-likelihood:

$$\mathcal{L}_{\text{AR}} = -\sum_{p \in \mathcal{P}} \sum_{i=1}^{|S_{\text{quantized}}^p|} \log P(s_i^p \mid s_1^p, \ldots, s_{i-1}^p),$$

where $s_i^p$ denotes the $i$-th token in $S_{\text{quantized}}^p$, and $P(s_i^p \mid \cdot)$ is the probability of predicting token $s_i^p$ given all preceding tokens.

### 3.4  Data sampling and postprocessing

Synthetic patient trajectories are generated by autoregressively sampling token sequences using the trained `TempoTransformer` with Top-$k$ sampling. Each synthetic sequence $\tilde{S}^p$ interleaves time tokens $\tau_i^p$ and event tokens $k_i^p$. Sampling is constrained to ensure structural integrity: the first $L_t$ tokens per block, representing time, are sampled from the time token vocabulary $[0, 9]$, while the subsequent $L_z \cdot D$ event tokens are sampled from the event token vocabulary $[K]$, with invalid token probabilities masked. Next, the event tokens are mapped to latent representations $\tilde{\mathbf{z}}_i^p$ via codebook $C$, decoded by `Dec` into text embeddings $\tilde{\mathbf{x}}_i^p$, and converted to text (*e.g.*, lab item Glucose value 95 uom mg/dL). Time tokens are detokenized into timestamps (e.g., $[7, 2] \rightarrow 720$ minutes). These text-based events and timestamps are then converted to relational tables.

Although the generated data typically reflect the original table and column names, we observed rare but noticeable issues such as misspelled table or column names or extraneous characters in numeric fields. We hypothesize these errors result from both the quantization step and the autoregressive token prediction step. To ensure accurate table construction, we apply the following postprocessing steps: **(1) Event-level verification**: Each event $e_i^p$ is validated to ensure that it begins with a valid table name and adheres to the *column-value* pair format. Misspelled column names are corrected by matching to the closest valid name using Levenshtein distance, and extraneous characters are removed from numeric fields. Only events that pass all checks after corrections are retained for further processing. **(2) Patient-level validation**: Subsequently, any sequence $S^p$ containing any invalid events $e_i^p$ are discarded. Retained sequences are checked for temporal consistency, removing events starting from any timestamps that do not follow chronological order or fall outside the observation window. Valid sequences are then converted into relational tables, followed by further postprocessing to enforce column-specific constraints (*e.g.*, numeric ranges). These steps collectively ensure that the final tables are structurally and semantically consistent with the expected format. Further details are provided in Appendix D.2, and the postprocessing algorithm is outlined in Algorithm 1.

## 4  Experiments

### 4.1  Setup

**Dataset**  In this study, we used two publicly available EHR datasets, MIMIC-IV [28] and eICU [29]. Adhering to the GenHPF [15] framework, which utilizes all EHR features for diverse prediction tasks, we focused on patients aged 18 years or older who had been admitted to the ICU for more than 12 hours. Based on this framework, we integrated three time-series tables (laboratory tests, prescriptions, and input events) from both datasets. To assess the generalizability of our approach

across diverse clinical settings, we conducted additional experiments on the MIMIC-IV-ED [30] emergency department dataset, with results reported in Appendix F.3. After preprocessing, the datasets comprised 62,704 patients (about 7 million rows) from MIMIC-IV and 143,812 (about 7 million rows) from eICU, with summary statistics detailed in Table 7. The dataset was split into a 9:1 ratio, allocating 90% for training and 10% for testing. The generative model was trained exclusively with the training data. We aimed to generate data across all columns to preserve the original dataset as comprehensively as possible; however, we excluded columns offering no analytical value (*e.g.*, *subject_id*, *orderid*). Details on exclusions are provided in Appendix B.3.

**Baseline**  This study is the first to synthesize raw EHR data, and therefore does not share an identical problem formulation with prior EHR synthesis research, making direct comparisons challenging. Because existing methods rely on predetermined feature selection and complex preprocessing steps, it is infeasible to extend them to the broader generative scope pursued in this work (*i.e.*, all elements in the original database structure). To provide a baseline, we adapt RealTabFormer [25], designed for text-based relational databases rather than time-series relational databases. Built on GPT-2, RealTabFormer is constrained by limited context length, restricting the amount of data it can process. To address this, we concatenate time-series data in chronological order and apply QLoRA [31] to Llama 3.1[32], rather than GPT-2 [33], allowing for longer sequences. Moreover, RealTabFormer does not compress data but instead models it directly, effectively serving as a *no-compression* baseline.

Additionally, we include baselines using SDV [34] (Gaussian Copula-based), RC-TGAN [35] (GAN-based), and ClavaDDPM [36] (diffusion-based) for multi-table generation. These methods, while not designed for time-series multi-table data, can generate synthetic data by modeling timestamps as standard columns within each table, as is typical in EHR database structures. However, these approaches might struggle to capture the temporal dynamics due to their lack of explicit temporal modeling. Appendix E.1 details the implementation and technical specifications.

## 4.2 Evaluation framework for synthetic multi-table time-Series EHRs

### 4.2.1 Existing evaluation frameworks

Evaluation of synthetic multi-table time-series data, particularly for raw EHRs, remains underexplored. Existing frameworks, such as SDMetrics[3] and Synthcity [37], primarily focus on single-table evaluation. In multi-table scenarios, these frameworks typically average metrics across individual tables or merge tables into a single one, obscuring inter-table relationships and temporal dynamics critical to EHR data. To our knowledge, no framework evaluates all components of raw multi-table EHR data while preserving its time-series structure. To address this, we propose an evaluation framework for `RawMed` that integrates standard metrics (*e.g.*, CDE), adapts metrics to handle the heterogeneity of raw EHRs (*e.g.*, I-CDE), and introduces novel temporal fidelity metrics (*e.g.*, Time Gap and Next Event Prediction). Detailed metric formulations are provided in the Appendix C.

### 4.2.2 Single-table evaluation

Single-table evaluation assesses the fidelity of synthetic data within individual tables, treating each row as an instance. To capture distributional similarities, we employ both low-order statistics and high-order metrics.

**Column-wise Density Estimation (CDE)** evaluates the similarity of marginal distributions for each column. For numeric columns, we use the Kolmogorov-Smirnov (KS) statistic (range: [0,1]), and for categorical columns, we use the Jensen-Shannon (JS) distance (range: [0,1]), both with lower values indicating higher similarity. In EHR data, a single column (*e.g.*, *amount* column) often contains measurements for diverse items (*e.g.*, about 2,000 drugs), identified by the item column (*e.g.*, *itemid* or *drug*). To address this, **Item-specific Column-wise Density Estimation (I-CDE)** filters data for specific items (e.g., creatinine), computes CDE per item, and averages the results across items to ensure the synthetic data preserves the unique characteristics of each clinical entity for precise item-level fidelity.

**Pairwise Column Correlation (PCC)** and **Item-specific Pairwise Column Correlation (I-PCC)** assess inter-column dependencies in synthetic data by comparing correlation matrices of real and

---

[3]https://docs.sdv.dev/sdmetrics

Table 2: Results of **single-table evaluation** on MIMIC-IV and eICU datasets. Metrics are averaged across columns and tables for each dataset. Lower values are better, with best in bold.

| | MIMIC-IV | | | | | | eICU | | | | | |
| | Column-wise ↓ | | Pair-wise ↓ | | High-order ↓ | | Column-wise ↓ | | Pair-wise ↓ | | High-order ↓ | |
| Model | CDE | I-CDE | PCC | I-PCC | ER | SMAPE | CDE | I-CDE | PCC | I-PCC | ER | SMAPE |
|---|---|---|---|---|---|---|---|---|---|---|---|---|
| Real | - | - | - | - | 15.35 | 60.85 | - | - | - | - | 44.05 | 43.42 |
| SDV | 0.11 | 0.54 | 0.26 | 0.26 | 49.32 | 103.85 | 0.13 | 0.61 | 0.19 | 0.19 | 76.34 | 101.97 |
| RC-TGAN | 0.26 | 0.54 | 0.18 | 0.28 | 38.21 | 97.26 | 0.34 | 0.59 | 0.18 | 0.21 | 69.08 | 111.25 |
| ClavaDDPM | 0.06 | 0.22 | 0.08 | 0.27 | 27.91 | 80.02 | 0.06 | 0.26 | 0.07 | 0.19 | 60.36 | 67.83 |
| RawMed | **0.04** | **0.05** | **0.04** | **0.10** | **19.69** | **57.31** | **0.05** | **0.08** | **0.06** | **0.10** | **45.58** | **48.42** |

synthetic tables. For PCC, matrices are computed using Pearson's correlation coefficient (range: $[-1, 1]$) for numeric-numeric pairs, Theil's U statistic (range: $[0, 1]$) for categorical-categorical pairs, and correlation ratio (range: $[0, 1]$) for categorical-numeric pairs. The mean absolute difference ($\mu_{abs}$) quantifies dependency fidelity by averaging element-wise matrix differences. I-PCC extends this by filtering data for each item, computing $\mu_{abs}$ per item's correlation matrix, and averaging across items for precise item-level fidelity.

For high-order dependencies, **Predictive Similarity** evaluates the ability of synthetic data to model complex, non-linear dependencies in single-table data through predictive performance. An XGBoost model is trained with each column as the target and remaining columns as inputs, with performance evaluated on real test data. For numeric targets, we use Symmetric Mean Absolute Percentage Error (SMAPE; range: [0,200]); for categorical targets, we use classification error rate (ER; range: [0,100]). A smaller performance gap between models trained on synthetic data compared to those trained on real data indicates effective capture of high-order dependencies.

### 4.2.3 Time-series multi-table evaluation

EHR data are typically stored across multiple tables, each containing time-series events for individual patients. To address the complex interactions among these tables, we preserve both the original structure and time-series aspects. Each patient identified by a primary key (*e.g.*, *stay_id*), is treated as an instance.

We evaluate the **Clinical Utility** of synthetic data for downstream predictive tasks in a multi-table EHR setting. Unlike single-table data, multi-table EHR data exhibit significant variability in event types and lengths per patient, necessitating comprehensive representations. To address this variability, we employ two methods: **GenHPF** [15], which lists all events sequentially as a single textual sequence, and **MEDS-TAB** [38], which aggregates events into fixed time intervals (see Appendix E.2.1 for details). For evaluation, we define 11 clinical prediction tasks (see Table 10), and report the Area Under the Receiver Operating Characteristic Curve (AUROC) for models trained on synthetic data compared to those trained on real data, using real test data.

We also perform **Membership Inference Attacks (MIA)** [39] to evaluate privacy leakage in synthetic data by measuring distances between synthetic samples and the training/test dataset to infer membership, where successful identification indicates inadequate privacy protection.

To evaluate the temporal fidelity, we introduce two metrics: **Time Gap** and **Event Count**. Time Gap quantifies the similarity of distributions of time gaps between consecutive events for each patient, employing the KS statistic (range: [0,1]) on timestamps. Event Count assesses the distribution of event counts per patient, also using the KS statistic, to evaluate whether synthetic data accurately reflects the frequency of clinical events.

Finally, **Next Event Prediction** evaluates temporal sequence dynamics using an LSTM [40]-based model to predict the next event's item or drug name, formulated as a multi-label classification task for concurrent events. The model is trained separately on synthetic and real data, with performance evaluated on real test data using the F1 score, quantifying the synthetic data's ability to capture event sequence patterns.

## 5 Results

In single-table evaluations, as presented in Table 2, RawMed demonstrates superior performance across CDE and PCC metrics. Notably, ClavaDDPM exhibits performance closer to RawMed compared to

Table 3: **Clinical utility evaluation** on MIMIC-IV and eICU datasets for downstream clinical tasks. Performance is measured by micro-averaged AUROC across 11 clinical prediction tasks (see Table 13 for individual task performance) under MEDS-TAB and GenHPF representations.

| | MIMIC-IV | | eICU | |
|---|---|---|---|---|
| Model | MEDS-TAB | GenHPF | MEDS-TAB | GenHPF |
| Real | 0.90±0.06 | 0.82±0.09 | 0.87±0.08 | 0.80±0.09 |
| SDV | 0.46±0.13 | 0.47±0.13 | 0.46±0.10 | 0.48±0.08 |
| RC-TGAN | 0.51±0.14 | 0.51±0.13 | 0.47±0.13 | 0.48±0.11 |
| ClavaDDPM | 0.68±0.19 | 0.64±0.17 | 0.66±0.14 | 0.63±0.12 |
| RawMed | **0.87**±0.08 | **0.80**±0.09 | **0.83**±0.10 | **0.78**±0.10 |

Table 4: **Temporal fidelity and privacy evaluation** on MIMIC-IV and eICU datasets. Next Event Prediction F1 scores are averaged over three random seed runs with std. The best results are in bold.

| | Next Event Predict (F1) ↑ | | Time Gap ↓ | | Event Count ↓ | | MIA (Accuracy) | |
|---|---|---|---|---|---|---|---|---|
| Model | MIMIC-IV | eICU | MIMIC-IV | eICU | MIMIC-IV | eICU | MIMIC-IV | eICU |
| Real | 0.18±0.06 | 0.30±0.00 | - | - | - | - | - | - |
| SDV | 0.05±0.01 | 0.13±0.00 | 0.76 | 0.50 | 0.46 | 0.36 | 0.499 | 0.502 |
| RC-TGAN | 0.02±0.00 | 0.07±0.01 | 0.43 | 0.39 | 0.04 | **0.03** | 0.500 | 0.500 |
| ClavaDDPM | 0.06±0.00 | 0.12±0.00 | 0.48 | 0.41 | 0.11 | 0.05 | 0.500 | 0.499 |
| RawMed | **0.16**±0.00 | **0.25**±0.03 | **0.01** | **0.03** | **0.02** | 0.05 | 0.498 | 0.497 |

SDV and RC-TGAN. However, in item-specific metrics, I-CDE and I-PCC, RawMed significantly outperforms ClavaDDPM. This underscores RawMed's capability to accurately preserve the characteristics of individual clinical items. Additionally, in Predictive Similarity, RawMed achieves the lowest SMAPE and ER among baselines, closely matching real data, demonstrating semantic consistency with the original dataset alongside robust I-CDE and I-PCC performance. For a detailed results on table-wise and column-wise fidelity, refer to Table 11 and 12.

In multi-table time-series evaluations, the performance disparity between RawMed and baseline methods becomes more pronounced. For Clinical Utility, Table 3 shows RawMed's AUROC scores closely match real data in downstream prediction tasks, surpassing ClavaDDPM, SDV, and RC-TGAN in both representations. Regarding temporal metrics, as shown in Table 4, RawMed excels in replicating inter-event interval distributions (Time Gap), achieving outstanding scores of 0.01–0.03 against baselines' 0.41–0.76. For Event Count, RawMed records competitive scores of 0.02 (MIMIC-IV) and 0.05 (eICU), though RC-TGAN outperforms it in eICU (0.03). RawMed also excels in Next Event Prediction, capturing event sequence patterns more effectively than other methods. Additionally, RawMed exhibits minimal vulnerability to membership inference attacks (MIA), performing near random guessing levels, thus ensuring robust privacy preservation.

Collectively, these results highlight RawMed's efficacy in generating multi-table time-series EHRs data that closely mirrors real data in terms of temporal precision, clinical utility, and privacy preservation. In contrast, baselines such as SDV, RC-TGAN, and ClavaDDPM underperform due to their lack of explicit temporal modeling and limited capability to model complex, large-scale EHR data.

**Compressed vs. Non-compressed approaches** We compared RealTabFormer (RTF, non-compressed) and RawMed (compressed) to evaluate the effect of sequence compression on synthetic data quality.[4] RTF models data in the text space, handling maximum sequence lengths of 11.2k for MIMIC-IV and 3.1k for eICU. In contrast, RawMed operates in the latent space, reducing these to 1.8k (84% reduction) and 0.8k (74% reduction), respectively. As reported in Table 5, RawMed outperformed RTF in most metrics, including time gap metrics of 0.05 (MIMIC-IV) and 0.01 (eICU) compared to RTF's 0.17 and 0.13, respectively. Thus, RawMed demonstrates superior data fidelity and temporal accuracy while significantly reducing sequence length.

---

[4]Experiments typically used a 12-hour observation window for training and synthesis, but both RTF and RawMed used a 6-hour window in this experiment due to RTF's context-size limits. We synthesized about 20k samples to match RTF-generated sample counts, as generating the original dataset size was infeasible for RTF.

Table 5: Results on MIMIC-IV and eICU datasets with a 6-hour observation window, comparing RealTabFormer (RTF) and `RawMed`. Lower is better, best in bold.

|  | MIMIC-IV | | eICU | |
|---|---|---|---|---|
| Metric | RTF | RawMed | RTF | RawMed |
| CDE ↓ | 0.11 | **0.04** | 0.21 | **0.06** |
| PCC ↓ | 0.08 | **0.03** | 0.19 | **0.09** |
| Time Gap ↓ | 0.17 | **0.05** | 0.13 | **0.01** |
| # Events ↓ | 0.20 | **0.07** | **0.07** | 0.08 |

Table 6: Ablation study on MIMIC-IV dataset. Variants exclude RQ (Residual Quantization), Time Tok. (Time Tokenization), and Time Sep. (Time Separation). Lower values are better, best in bold.

| Variants | CDE | PCC | Time Gap | # Events |
|---|---|---|---|---|
| **Full Method** | **0.04** | **0.04** | **0.01** | **0.02** |
| w/o RQ | 0.05 | 0.04 | 0.05 | 0.13 |
| w/o Time Tok. | 0.04 | 0.04 | 0.04 | 0.12 |
| w/o Time Sep. | 0.07 | 0.07 | 0.51 | 0.40 |

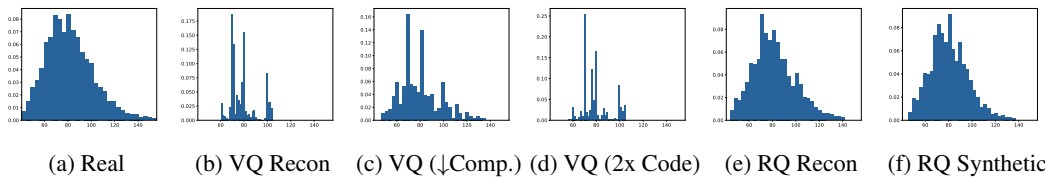

| (a) Real | (b) VQ Recon | (c) VQ (↓Comp.) | (d) VQ (2x Code) | (e) RQ Recon | (f) RQ Synthetic |
|---|---|---|---|---|---|

Figure 2: Comparison of VQ-VAE and RQ-VAE for the *patientweight* column in MIMIC-IV. Subfigures: (a) real data, (b) VQ reconstruction, (c) VQ with less compression, (d) VQ with doubled codebook, (e) RQ reconstruction, (f) RQ synthetic data. Shared x-axes with proportional y-axes.

## 6 Ablation Studies

**RQ vs. VQ: Why RQ is more suitable** Although Vector Quantization (VQ) offers robust compression within the `RawMed` framework, it does not consistently achieve high-fidelity representation across all data types. In text-based EHR data, columns with low correlation and independent distributions (e.g., *patientweight*) are challenging to encode with VQ's limited codebook, resulting in significant distortion of the original distribution upon reconstruction, as shown in Figure 2(b). Halving the compression ratio (Figure 2(c)) or doubling the codebook size (Figure 2(d))—note that these data are reconstructed, not generated—fails to fully resolve distortion in these numerical columns. In contrast, `RawMed` employs Residual Quantization (RQ) for multi-stage quantization, effectively preserving independent column distributions (Figure 2(e)–(f)). Table 14 shows that for *patientweight* in MIMIC-IV, VQ's KS statistic (0.28) far exceeds RQ's (0.09), confirming VQ's greater distortion.

**Ablation study of `RawMed` components** Table 6 evaluates `RawMed` variants with components removed, with lower metric values indicating better performance. The full method achieves optimal results across all metrics. Removing RQ (*i.e.*, using VQ) degrades performance, increasing CDE to 0.05, Time Gap to 0.05, and Event Count to 0.13. Excluding Time Tokenization, which decomposes timestamps into digits with a 0–9 vocabulary, impairs time pattern recognition, raising Time Gap to 0.04 and Event Count to 0.12. Omitting Time Separation, which isolates timestamps for explicit temporal modeling, causes the largest performance drop, elevating CDE to 0.07, Time Gap to 0.51, and Event Count to 0.40. These results highlight the critical role of each component in `RawMed`'s temporal modeling efficacy.

**Scalability** In this study, we conducted experiments primarily with a 12-hour observation window, also testing 6-hour and 24-hour windows to evaluate `RawMed`'s scalability. As shown in Table 15, most metrics remained stable across windows, though the KS statistics for the event count increased slightly for the 24-hour window. This demonstrates `RawMed`'s potential to generalize across different time scales.

## 7 Discussion

This study introduces `RawMed`, a framework for synthesizing multi-table time-series EHR data while retaining all columns and original values. By enabling hospitals to generate high-fidelity synthetic EHRs with minimal preprocessing and providing users with realistic data for flexible downstream tasks, `RawMed` can accelerate medical AI research while safeguarding patient privacy. However,

several limitations remain with respect to fully encompassing the complexity of real-world medical settings. First, this work focuses on three primary tables, which leaves the question whether it can handle a significantly higher number (*e.g.*, dozens) of tables. In addition, extended time-series data might require advanced compression or sampling methods. While the framework could potentially incorporate conditional generation (including static attributes such as gender or birth year), this study, as an initial approach, focuses on unconditional generation, leaving these specialized applications for future research. In subsequent work, we plan to expand the range of tables, implement conditional generation, and more fully integrate static features into the model design, thereby providing more realistic synthetic EHR data for a broader array of medical AI tasks.

## Acknowledgments and Disclosure of Funding

This work was supported by the Institute of Information & Communications Technology Planning & Evaluation (IITP) grants (No.RS-2019-II190075), the Korea Health Industry Development Institute (KHIDI) grant (No.HR21C0198, No.RS-2025-02213750) and the National Research Foundation of Korea (NRF) grants (NRF-2020H1D3A2A03100945), funded by the Korea government (MSIT, MOHW).

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

# A  Feature definitions in EHR data

## A.1  Feature definition

In electronic health record (EHR) data, tables (*e.g.*, *lab*, *medication*, *infusion*) contain **columns**, some of which are **item columns** (e.g., *itemid*, *drug*, *labname*) that identify clinical events, such as which lab tests or medication administration. A **feature** is a processed representation of data derived from one or more columns, typically created by selecting, combining, or transforming column data, as described in the Introduction.[1]

To define a feature from EHR data, the following steps are typically applied:

- **Select a specific item:** A specific item is selected to represent a clinical event. For example, in a laboratory table, selecting *itemid* = 12345 might isolate all instances of a "blood glucose" test.

- **Incorporate relevant categorical columns:** Related categorical columns (e.g., *uom*, *dose_unit*, or *route*) further refine the feature definition. For instance, two rows with the same *itemid* but different *uom* values, such as "mg/dL" vs. "mmol/L", would represent distinct features.

- **Include numerical columns:** Numerical columns (*e.g.*, *valuenum*, *amount*, or *rate*) provide the actual measurement or dosage values for the item. Typically, a feature can be defined by combining an item, a categorical column (e.g., *uom*), and a numerical column's values. Each numerical column may also represent a distinct feature when it captures a unique aspect of a clinical event, such as a dosage rate versus a total amount.

For example, "blood glucose level" is defined by *valuenum* where *itemid* = 12345 and *uom* = "mg/dL". While feature engineering is commonly used, this section focuses on the core structure of features, excluding feature engineering techniques (*e.g.*, aggregation or transformation).

## A.2  Feature count calculation

To calculate the number of features in EHR data, we define two approaches: the theoretical maximum (**Possible Feature Count**) and the practical estimate (**Actual Feature Count**). The data consists of a set of tables $\mathcal{E}$, such as medication, laboratory, or infusion tables. Each table $\epsilon \in \mathcal{E}$ includes categorical columns $C_{\epsilon,\mathrm{cat}}$ and numerical columns $C_{\epsilon,\mathrm{num}}$.

The following definitions are essential for calculating feature counts:

- **Distinct Items** ($I_\epsilon$): The set of unique items in table $\epsilon$, representing distinct clinical event identifiers in an item column. For example, in a medication table, each item $i \in I_\epsilon$ corresponds to a drug name (*e.g.*, paracetamol, aspirin).

- **Number of Unique Values in a Categorical Column** ($|V_{i,c}|$): The count of unique values in categorical column $c \in C_{\epsilon,\mathrm{cat}}$ (e.g., administration route) for item $i$. For instance, if paracetamol has administration routes "intravenous" and "oral," then $|V_{i,\mathrm{route}}| = 2$.

- **Number of Unique Categorical Value Combinations** ($|C_{\epsilon,\mathrm{cat}}^{\mathrm{unique}}(i)|$): The number of observed combinations of all categorical columns values for item $i$. For example, if paracetamol has three combinations of route and dose unit, then $|C_{\epsilon,\mathrm{cat}}^{\mathrm{unique}}(i)| = 3$.

- **Number of Numerical Columns** ($|C_{\epsilon,\mathrm{num}}|$): The count of numerical columns (e.g., amount, rate) associated with each item-category combination.

**Possible Feature Count**   The Possible Feature Count assumes all possible combinations of categorical values exist for each item, providing a theoretical upper bound. It is calculated as:

$$\text{Possible Feature Count} = \sum_{\epsilon \in \mathcal{E}} \left( \sum_{i \in I_\epsilon} \left( \prod_{c \in C_{\epsilon,\mathrm{cat}}} |V_{i,c}| \right) \times (|C_{\epsilon,\mathrm{num}}| + 1) \right),$$

where $\prod_{c \in C_{\epsilon,\mathrm{cat}}} |V_{i,c}|$ represents the number of possible categorical combinations, and $|C_{\epsilon,\mathrm{num}}| + 1$ accounts for the numerical columns plus a indicator feature representing the presence or absence of the categorical combination.

**Actual Feature Count**   The Actual Feature Count considers only the categorical combinations observed in the data, reflecting dependencies or sparsity that reduce the number of combinations. It is calculated as:

$$\text{Actual Feature Count} = \sum_{\epsilon \in \mathcal{E}} \left( \sum_{i \in I_\epsilon} \left| C_{\epsilon,\text{cat}}^{\text{unique}}(i) \right| \times (|C_{\epsilon,\text{num}}| + 1) \right),$$

where $\left| C_{\epsilon,\text{cat}}^{\text{unique}}(i) \right|$ is the number of actual combinations, and $|C_{\epsilon,\text{num}}| + 1$ accounts for numerical columns and the presence/absence feature.

For example, consider a medication table with 100 unique medications (*drug*), two categorical columns (*dose_unit* with 5 unique values and *route* with 3 unique values), and two numerical columns (*amount* and *rate*). If all $5 \times 3 = 15$ combinations of *dose_unit* and *route* are valid, the features per medication are calculated as $15 \times (2 + 1) = 45$, resulting in a total of $100 \times 45 = 4{,}500$ features. However, if only 10 combinations of *dose_unit* and *route* are present, the features per medication decrease to $10 \times (2 + 1) = 30$, and the total feature count reduces to $100 \times 30 = 3{,}000$.

In summary, the Possible Feature Count provides a theoretical maximum, while the Actual Feature Count reflects the data's sparsity and structure for a realistic estimate. This study used the Actual Feature Count to calculate the number of generated features (Table 1).

## B   Dataset statistics

### B.1   Dataset selection criteria

To synthesize time-series electronic health records (EHR), we selected ICU patient datasets, specifically **MIMIC-IV** and **eICU**, for their high temporal resolution and continuous monitoring. MIMIC-IV, collected from Beth Israel Deaconess Medical Center, includes 94,458 ICU stays across 364,627 patients, while eICU, a multi-center dataset, comprises 200,859 ICU stays from 139,367 patients across U.S. hospitals. These datasets provide large cohorts and diverse clinical variables, making them established benchmarks for validating most prior EHR synthesis studies. We focused on time-series tables, *labevents*, *prescriptions*, and *inputevents* from MIMIC-IV, and *lab*, *medication*, and *infusiondrug* from eICU. These tables capture critical temporal changes in physiological states, treatment interventions, and drug administration, enabling the synthesis of complex clinical patterns. In contrast, emergency department (ED) and outpatient datasets, with irregular intervals and lower resolution, are typically less suitable for synthesizing the continuous temporal dynamics of patient trajectory.

However, to evaluate the model's generalizability, we additionally utilized the **MIMIC-IV-ED** dataset to validate its ability to synthesize data in a distinct clinical setting. From MIMIC-IV-ED, we synthesized the *vitalsign* (time-series data such as heart rate and blood pressure), *medrecon* (static data on medications at admission), and *pyxis* (event-based data on medication dispensing via the Pyxis system) tables. By simultaneously generating time-series (*vitalsign*, *pyxis*) and non-time-series (*medrecon*) data, the model effectively synthesized complex patterns in ED data, demonstrating robustness across diverse clinical scenarios.

### B.2   Data statistics

This study utilized the publicly available MIMIC-IV and eICU datasets, focusing on patients aged 18 years or older with ICU stays exceeding 12 hours. Key metrics, including patient counts, event frequencies, and row counts for the tables, are summarized in Table 7 Additionally, ablation studies exploring 6- and 24-hour observation windows for MIMIC-IV and eICU, and a 6-hour window for the MIMIC-IV-ED dataset in ED settings, are detailed in Table 8.

### B.3   Criteria for excluded columns

In this study, we aimed to utilize as many columns as possible but excluded specific columns for the following reasons and generated all others. First, we removed patient or prescription identifiers and unique hospital-specific codes (*e.g.*, *subject_id*, *orderid*, *pharmacy_id*) that were not directly relevant to the clinical information used by the model.

Table 7: Summary statistics for preprocessed **MIMIC-IV** and **eICU** datasets with a 12-hour observation window, including patient counts, event frequencies, and row and column counts for lab (*labevents* or *lab*), medication (*prescriptions* or *medication*), and input (*inputevents* or *infusiondrug*) data. Note: M denotes million.

| Metric | MIMIC-IV | eICU |
|---|---|---|
| Number of Patients | 62,704 | 143,812 |
| Max. Events per Patient | 243 | 114 |
| Avg. Events per Patient | 108.8 | 47.2 |
| Total Rows | 6.8M | 6.8M |
| Rows: Lab | 3.8M | 4.6M |
| Rows: Medication | 1.7M | 1.6M |
| Rows: Input | 1.4M | 0.6M |
| Columns: Lab | 9 | 6 |
| Columns: Medication | 9 | 6 |
| Columns: Input | 15 | 6 |

Table 8: Summary statistics for preprocessed **MIMIC-IV-ED** (6-hour observation window), **MIMIC-IV**, and **eICU** (6- and 24-hour observation windows) datasets. Note: M denotes million.

| Metric | MIMIC-IV-ED | MIMIC-IV (6h) | MIMIC-IV (24h) | eICU (6h) | eICU (24h) |
|---|---|---|---|---|---|
| Observation Window (hrs) | 6 | 6 | 24 | 6 | 24 |
| Number of Patients | 144,790 | 63,903 | 51,357 | 132,202 | 119,641 |
| Max. Events per Patient | 34 | 165 | 366 | 79 | 179 |
| Avg. Events per Patient | 16.5 | 71.6 | 170.2 | 32.0 | 74.1 |
| Total Rows | 2.4M | 4.6M | 8.7M | 4.2M | 8.9M |

Second, we excluded columns that contained multiple timestamps or offset columns conveying similar time information (*e.g.*, *charttime*, *starttime*). While each timestamp could potentially have clinical value, we consolidated them into a single representative time field to reduce analytical complexity.

Finally, we excluded columns that potentially contained future information beyond the event time to avoid overestimating model performance by training on data not available at the actual prediction point. All remaining columns were included in the modeling process. Table 9 summarizes all excluded columns and indicates the primary reason for each exclusion.

## C   Evaluation framework details

To rigorously evaluate the quality of synthetic multi-EHR data generated by `RawMed`, we define a comprehensive set of metrics that assess both single-table fidelity and multi-table temporal dynamics. Each metric is carefully selected to capture specific aspects of data quality, such as distributional similarity, inter-column dependencies, temporal fidelity, clinical utility, and privacy. Below, we provide detailed definitions, mathematical formulations, and justifications for each metric, supported by visualizations where applicable. Our evaluation pipeline is designed to be robust and tailored to the unique challenges of raw multi-table time-series EHR data, providing a thorough assessment of synthetic data quality.

### C.1   Single-table evaluation

Single-table evaluation focuses on the fidelity of synthetic data within individual tables, treating each row as an independent instance. We combine low-order and high-order metrics to capture marginal distributions and complex inter-column dependencies. Below, we detail each metric, its formulation, and its relevance to EHR data evaluation.

- **Column-wise Density Estimation (CDE)**: CDE measures the similarity of distributions between real and synthetic data for each column. For numeric columns, we use the Kolmogorov-Smirnov (KS) statistic, defined as:

$$\text{KS} = \sup_{x} |F_r(x) - F_s(x)|,$$

Table 9: Columns excluded from each dataset table, categorized by the reason for exclusion: identifiers or codes, time-related columns, or columns containing potential future information.

| Dataset | Table | Excluded Columns |
|---------|-------|------------------|
| *Reason: Identifiers or Codes* | | |
| MIMIC-IV | labevents | labevent_id, subject_id, specimen_id, order_provider_id |
| MIMIC-IV | inputevents | subject_id, caregiver_id |
| MIMIC-IV | prescriptions | subject_id, pharmacy_id, poe_id, poe_seq, order_provider_id, orderid, linkorderid |
| eICU | lab | labid |
| eICU | infusiondrug | infusiondrugid |
| eICU | medication | medicationid, drughiclseqno, gtc |
| *Reason: Time Columns* | | |
| MIMIC-IV | labevents | storetime |
| MIMIC-IV | inputevents | endtime, storetime |
| MIMIC-IV | prescriptions | stoptime |
| eICU | lab | labresultrevisedoffset |
| eICU | medication | drugorderoffset, drugstopoffset |
| *Reason: Future Information Leakage* | | |
| MIMIC-IV | prescriptions | statusdescription |
| eICU | medication | drugordercancelled |

where $F_r(x)$ and $F_s(x)$ are the cumulative distribution functions (CDFs) of the real and synthetic data, respectively. The KS statistic ranges from $[0, 1]$, with lower values indicating higher similarity. For categorical columns, we use the Jensen-Shannon (JS) divergence, defined as:

$$\mathrm{JS}(P_r\|P_s) = \frac{1}{2}\mathrm{KL}(P_r\|M) + \frac{1}{2}\mathrm{KL}(P_s\|M),$$

where $P_r$ and $P_s$ are the probability distributions of the real and synthetic data, $M = \frac{1}{2}(P_r + P_s)$, and KL is the Kullback-Leibler divergence. JS ranges from $[0, 1]$, with lower values indicating greater similarity.

– **Justification**: CDE is a fundamental metric for comparing distributional similarity across tabular data. It effectively captures the distributional properties of numerical columns (e.g., laboratory values) and categorical columns (e.g., prescribed drug names) in EHR data. This ensures robust evaluation across diverse column types, critical for synthetic data validity.

• **Item-specific Column-wise Density Estimation (I-CDE)**: I-CDE extends CDE by evaluating distributional fidelity for specific clinical items (e.g., creatinine, glucose) within columns that aggregate multiple item types (e.g., a *value* column linked to an *itemid*). For each item, we filter the data and apply the same KS or JS metrics as in CDE. To ensure robustness, we exclude items with fewer than 0.1% of total records (*e.g.*, lab tests occurring 1–2 times in 3.8 million events) to avoid skewed results due to extremely rare events.

– **Justification**: I-CDE addresses the heterogeneity of EHR data, where a single column may represent diverse clinical entities (*e.g.*, 2,000 drugs). By ensuring synthetic data preserves item-specific distributions, I-CDE enhances clinical validity.

• **Pairwise Column Correlation (PCC)**: PCC assesses the fidelity of inter-column dependencies by comparing correlation matrices of real and synthetic data. For numeric-numeric pairs, we use Pearson's correlation coefficient ($\rho \in [-1, 1]$). For categorical-categorical pairs, we use Theil's U statistic ($U \in [0, 1]$), which measures conditional entropy. For categorical-numeric pairs, we use the correlation ratio ($\eta \in [0, 1]$), which quantifies the variance explained by the categorical variable. The mean absolute difference ($\mu_{\mathrm{abs}}$) between real and synthetic correlation matrices is computed as:

$$\mu_{\mathrm{abs}} = \frac{1}{N} \sum_{i,j} |\mathrm{Corr}_r(i, j) - \mathrm{Corr}_s(i, j)|,$$

where $\mathrm{Corr}_r$ and $\mathrm{Corr}_s$ are the correlation matrices for real and synthetic data, and $N$ is the number of matrix elements.

- **Justification**: PCC captures dependencies between clinical variables (e.g., heart rate and blood pressure), essential for realistic synthetic EHR data. The use of multiple correlation measures ensures applicability to mixed data types in EHRs.

- **Item-specific Pairwise Column Correlation (I-PCC)**: I-PCC extends PCC by computing correlation matrices for each item-specific subset of the data (e.g., all records for a specific *itemid*). The $\mu_{\mathrm{abs}}$ is calculated for each item's matrix and averaged across items. Items with fewer than 100 records are excluded.

  - **Justification**: I-PCC evaluates item-specific inter-column dependencies, ensuring that synthetic data captures relationships within subsets of EHR data (*e.g.*, correlations between dosage and frequency for a specific drug). This is critical for maintaining clinical relevance.

- **Predictive Similarity**: Predictive Similarity evaluates the ability of synthetic data to capture high-order, non-linear dependencies by training an XGBoost model to predict each column using the remaining columns as features. Models are trained on synthetic data and evaluated on real test data. For numeric targets, we use Symmetric Mean Absolute Percentage Error (SMAPE), defined as:

$$\mathrm{SMAPE} = \frac{100}{n} \sum_{i=1}^{n} \frac{|y_i - \hat{y}_i|}{(|y_i| + |\hat{y}_i|)/2},$$

where $y_i$ and $\hat{y}_i$ are the real and predicted values, and SMAPE ranges from [0,200]. For categorical targets, we use the classification error rate (ER), defined as:

$$\mathrm{ER} = \frac{1}{n} \sum_{i=1}^{n} \mathbb{I}(y_i \neq \hat{y}_i) \times 100,$$

where ER ranges from [0,100]. A smaller performance gap between models trained on synthetic versus real data indicates better capture of dependencies.

  - **Justification**: Predictive similarity captures complex relationships that low-order metrics may not detect, indirectly assessing semantic consistency. While Mean Squared Error (MSE) and Mean Absolute Error (MAE) are commonly used regression metrics, their scale-dependency led to the adoption of SMAPE, which is bounded between 0 and 200 for consistent evaluation across varying ranges.

### C.2 Time-series multi-table evaluation

Multi-table EHR data involve time-series events across multiple tables, linked by a primary key (e.g., *stay_id*). Our evaluation metrics preserve this structure and assess temporal fidelity, clinical utility, and privacy. Below, we detail each metric and its rationale.

- **Clinical Utility**: To assess clinical utility, we evaluate the performance of synthetic data on downstream predictive tasks. We defined 11 clinical prediction tasks (e.g., predicting creatinine and hemoglobin levels), following the task definitions from [15]. Details are provided in Table 10. To address variability in the number and types of events across patients, we employ two representation methods, GenHPF and MEDS-TAB, as detailed in E.2.1. For each task, the synthetic data is split into two segments over an observation window of length $T$ (*e.g.*, $T = 12$ hours. The first half (*e.g.*, the first 6 hours) serves as input for the predictive model, with predictions made at the midpoint ($T/2$). For multi-class tasks, the label is the last value of the lab measurement in the second half (*e.g.*, the subsequent 6 hours). For binary tasks, the label indicates whether a medication event occurs in the second half. Models are trained on synthetic data and tested on real data, with performance measured using the Area Under the Receiver Operating Characteristic Curve (AUROC). Higher AUROC scores indicate better utility for clinical applications.

  - **Justification**: Clinical utility reflects the practical value of synthetic data for real-world EHR applications. Using two representation methods ensures robustness across diverse modeling approaches.

- **Membership Inference Attacks (MIA)**: MIA assesses privacy leakage by attempting to infer whether a sample was part of the training dataset. We compute distances (e.g., Hamming distance) between synthetic samples and the real training set, using a threshold-based classifier to predict membership. The attack's success rate (accuracy) indicates privacy risk, with lower values reflecting better privacy protection. This is conducted using the GenHPF representation.
  - **Justification**: MIA is a standard metric for evaluating privacy in synthetic EHR data, as it measures the risk of re-identifying individuals from the training set, a critical concern given EHRs' sensitive patient information.

- **Time Gap**: Time Gap evaluates the similarity of time interval distributions between consecutive patient events, using the KS statistic. In EHR data, simultaneous events (zero intervals) are common due to concurrent clinical recordings in clinical workflows or temporal resolution constraints of EHR systems. To address this, we compute KS statistics both including and excluding zero intervals, with results excluding zero intervals reported in Table 4. CDFs for both cases are visualized in Figure 4: panels (a)–(b) show distributions including simultaneous events, and panels (c)–(d) show distributions excluding them. Additionally, absolute event time distributions (from admission to event occurrence) are presented in Figure 4(e)–(f).
  - **Justification**: Time Gap ensures synthetic data accurately captures the timing of clinical events, essential for realistic patient trajectories in time-sensitive settings like ICUs. By evaluating both zero and non-zero intervals, it accounts for EHR-specific recording patterns, while the KS statistic provides a robust, non-parametric measure of temporal fidelity.

- **Event Count**: Event Count evaluates the similarity in the distribution of the number of clinical events per patient between synthetic and real EHR data, employing the (KS statistic to compare these distributions. The CDFs of event counts for both synthetic and real data are visualized in Figure 4(g)–(h).
  - **Justification**: Event Count verifies that synthetic data replicates the frequency of clinical events per patient, a key characteristic of patient trajectories.

- **Next Event Prediction**: Next Event Prediction evaluates the synthetic data's ability to capture temporal sequence dynamics. An LSTM-based model [40] predicts the next event's item or drug name, formulated as a multi-label classification task to handle concurrent events. The model is trained on synthetic data and evaluated on real test data using the F1 score, which balances precision and recall for multi-label predictions. Higher F1 scores indicate better sequence modeling.
  - **Justification**: This metric captures the sequential patterns of clinical events, essential for modeling patient trajectories. The LSTM's ability to model long-term dependencies suits the complex temporal structure of EHRs.

### C.3 Visualizations and results

To provide a comprehensive view of synthetic data quality, we include the following figures and tables:

- **Figure 3**: Correlation matrix heatmap showing differences in Pairwise Column Correlation (PCC) between real and synthetic data.
- **Figure 4**: Cumulative Distribution Function (CDF) plots for Time Gap (including and excluding zero intervals), absolute event time, and event count distributions.
- **Table 2**: Overall single-table evaluation results, averaged across tables and columns for each dataset.
- **Tables 11, 12**: Column-wise evaluation results, including CDE, I-CDE, SMAPE, ER gaps for predictive similarity, and null ratio of real and synthetic data.
- **Table 3**: AUROC comparisons for clinical utility tasks across datasets.
- **Table 10**: AUROC results for clinical utility tasks, detailing performance for each task.

- **Table 4**: Accuracy for Membership Inference Attacks (MIA), F1 scores for Next Event Prediction, and KS statistics for Time Gap and Event Count.

By combining rigorous metric definitions, mathematical formulations, and extensive visualizations, our evaluation pipeline offers a thorough and principled approach to assessing synthetic multi-table EHR data. This comprehensive framework not only addresses existing gaps in evaluation methodologies but also sets a new standard for evaluating time-series data in healthcare applications.

# D  RawMed framework details

## D.1  Architecture details

This section details the architectural components of the `RawMed` framework, consisting of two distinct modules: event compression and inter-event temporal modeling, providing specifics beyond the main text.

The event compression module transforms serialized clinical event text into compact latent representations, leveraging a Vector Quantized Variational AutoEncoder (VQ-VAE) [27] or Residual Quantization (RQ) framework [16], both implemented using 1D convolutional neural networks (CNNs). The processing pipeline begins with the construction of input embeddings, followed by encoding, quantization, and decoding stages.

The input embedding, denoted $\mathbf{x}_i^p \in \mathbb{R}^{L \times F}$, integrates three components adapted from an established text-based EHR prediction framework, GenHPF, to ensure robust representation of clinical data. The textual embedding ($\mathbf{x}_{\text{text},i}^p$) employs framework-derived embeddings for serialized event text, such as "lab item Glucose value 95 uom mg/dL", tokenized using a Bio+Clinical BERT tokenizer [41]. The type embedding ($\mathbf{x}_{\text{type},i}^p$) assigns categorical labels to tokens, distinguishing table names, column names, and values to capture the relational structure of the data. The digit-place embedding ($\mathbf{x}_{\text{dpe},i}^p$) enhances numeric tokens by tokenizing digits with spaces (e.g., "123.45" as "1 2 3 . 4 5") and assigning position-specific types for integer and decimal places. These components are combined as $\mathbf{x}_i^p = \mathbf{x}_{\text{text},i}^p + \mathbf{x}_{\text{type},i}^p + \mathbf{x}_{\text{dpe},i}^p$. The main text adopts the notation $\mathbf{x}_i^p = \mathbf{x}_{\text{text},i}^p$ for simplicity, though the full embedding incorporates type and digit-place embeddings to enhance structural and numeric fidelity.

The encoder (`Enc`) processes the input embedding through five CNN layers, each with a kernel size of 5, stride of 2, and padding of 2, followed by batch normalization and ReLU activation. This configuration ensures a receptive field sufficient to encompass the entire event sequence of length $L = 128$, capturing contextual dependencies. The encoder outputs a latent representation, $\hat{\mathbf{z}}_i^p \in \mathbb{R}^{L_z \times F_z}$, which is subsequently quantized.

The decoder (`Dec`) reconstructs the text, type, and digit-position embeddings ($\hat{\mathbf{x}}_{\text{text},i}^p, \hat{\mathbf{x}}_{\text{type},i}^p, \hat{\mathbf{x}}_{\text{dpe},i}^p$) using transposed CNN layers. The training objective minimizes reconstruction losses for all three embedding components, ensuring faithful recovery of text, type, and digit-position information.

The temporal modeling module, implemented as `TempoTransformer`, is a Transformer-based architecture for autoregressive prediction of quantized event sequences interleaved with tokenized timestamps. The model employs causal attention to ensure predictions depend only on prior tokens. Positional encodings provide sequence order information, and the model is trained from scratch, as the autoregressive target is the quantized codebook, obviating the need for fine-tuning from pretrained weights.

## D.2  Postprocessing

This appendix elaborates on the two-stage postprocessing pipeline applied to convert synthetic patient trajectories into relational tables, aligning with the structural and semantic consistency of real data. The first stage, **converting text to tabular format**, involves the core steps of event-level verification and patient-level validation (Algorithm 1). The second stage, **enhancing data quality**, applies additional column-specific constraints to refine the synthetic tables' statistical fidelity, as referenced in Section 3.4.

**Column-Specific Constraint Enforcement**    After validated sequences are converted into relational tables, further postprocessing ensures compliance with column-specific constraints, addressing potential errors from the generative process (e.g., quantization or autoregressive sampling). The following steps are applied:

- **Numeric Columns**: Values are filtered to lie within the minimum and maximum ranges observed in the real data, applied at both the column and item levels. Non-compliant events (e.g., a glucose value outside the real data's range) are removed.

- **Categorical Columns**: Invalid values not present in the permissible values for each column are replaced with the closest valid value, determined using Levenshtein distance similarity. A distance threshold ensures that only sufficiently similar replacements are accepted; otherwise, the value is flagged as invalid.

- **Patient-Level Filtering**: If any events contains a numeric value outside the valid range or a categorical value exceeding the Levenshtein distance threshold, the entire patient sample associated with that row is removed to maintain data integrity.

These steps ensure that the final relational tables adhere to the structural and statistical characteristics of the real data, removing outliers and ensuring both numerical and categorical validity.

**Modification and Rejection Rate Analysis**    To quantify the impact of our validation protocols, we analyzed the modification and rejection rates across both postprocessing stages for the eICU and MIMIC-IV datasets.

During the initial stage, **converting text to tabular format**, $0.42\%$ of events were modified and $0.07\%$ were rejected in eICU. For MIMIC-IV, these rates were $0.73\%$ and $0.17\%$, respectively. At the patient level, however, rejection rates were notably higher, reaching $2.71\%$ for eICU and $18.23\%$ for MIMIC-IV.

In the subsequent stage, **enhancing data quality**, stricter column-specific constraints were enforced. This led to event-level rates of $1.39\%$ modified and $0.49\%$ rejected for eICU, and $1.07\%$ modified and $0.79\%$ rejected for MIMIC-IV. These stringent checks led to a substantial increase in patient-level rejection rates, which rose to $16.90\%$ in eICU and $54.53\%$ in MIMIC-IV.

The pronounced disparity between patient-level and event-level rejection rates is a direct consequence of our stringent data integrity policy. A single erroneous event necessitates the rejection of the entire patient trajectory. This effect is particularly notable for datasets with long sequences, (e.g., with hundreds of events), all of which must be accurate for the trajectory to be retained. We emphasize that this full post-processing is crucial for `RawMed`'s data integrity. We do not recommend partial post-processing, as it fundamentally compromises the quality of the synthetic data.

# E    Training details and hyperparameters

## E.1    Generative models

- **RawMed** The framework employs a two-stage training process: the event compression module is trained first to generate quantized representations, followed by the inter-event temporal modeling module to predict sequences autoregressively.

    - **Event-level compression**: The event compression module uses the AdamW optimizer (learning rate: 5e-4, weight decay: 0.01), processing batches of 4096 events for up to 200 epochs, with early stopping after 10 epochs of stagnant validation accuracy. The loss function combines reconstruction losses for text, type, and digit-place embeddings with a commitment loss (commitment cost: 1.0). An EMA decay factor of 0.8 is applied for codebook updates in VQ-VAE training. A dropout rate of 0.2 is used for regularization. The input embedding has sequence length $L = 128$ and embedding dimension $F = 256$, compressed to a latent representation with $L_z = 4$ and $F_z = 256$. The codebook contains $K = 1024$ entries, with residual quantization (RQ) using a depth of $D = 2$, yielding a $4 \times 2$ code representation. Training is performed on a single NVIDIA A6000 GPU, completing in under 24 hours.

- **Temporal modeling between events**: The temporal modeling module uses the AdamW optimizer (learning rate: 3e-4, weight decay: 0.01), with batches of 32 sequences for up to 200 epochs, with early stopping after 10 epochs without improvement. The `TempoTransformer` consists of 12 layers, each with 8 attention heads, a hidden dimension of 512, and a feed-forward dimension of 2048. The input sequence includes time tokens ($\tau_i^p \in \{0, \ldots, 9\}^2$, length 2), event tokens ($k_i^p \in [1024]^{4 \times 2}$, codebook size 1024), and special tokens (<SOS>, <EOS>, <PAD>), resulting in a vocabulary size of 1037. Training was conducted on three NVIDIA A6000 GPUs for MIMIC and on a single NVIDIA A6000 GPU for eICU, both completing in under 48 hours.

  - **Generation**: Data generation uses top-$k$ sampling with $k = 250$ for MIMIC-IV and $k = 150$ for eICU.

- **RealTabFormer** [25]: Text-based autoregressive model for generating relational database. This model involves fine-tuning the Meta-LLaMA-3.1-8B model and a generation step using top-p sampling.

  - **Fine-tuning**: The Meta-LLaMA-3.1-8B model [32] was fine-tuned using QLoRA [31] (rank $r$=128, LoRA scaling factor $\alpha$=128, LoRA dropout=0.05) and Flash Attention-2 for memory optimization. Training used a maximum sequence length of 11,240 tokens, matching the input data. The Adafactor optimizer was applied with a learning rate of 2e-4, weight decay of 0.001, a batch size of 1, gradient accumulation (step size=4), and gradient checkpointing. Training ran for 2 epochs on a single RTX 3090 GPU, taking approximately 10 days for MIMIC-IV and 20 days for eICU.

  - **Generation**: Used the fine-tuned Meta-LLaMA-3.1-8B model with top-p sampling, setting the temperature to 1 and a threshold of 0.7.

- **SDV** [34]: The Synthetic Data Vault (SDV) uses the Hierarchical ML Algorithm (HMA) Synthesizer to generate synthetic relational data, modeling individual tables with Gaussian Copulas while preserving parent-child relationships.The SDV framework's default hyperparameters are applied for training.

- **RC-TGAN**: [35] A GAN-based model designed to generate synthetic relational databases by modeling parent-child relationships.

  - **Training and generation**: Uses default hyperparameters from the official repository, with batch size set to 500,000, the maximum feasible value for the NVIDIA RTX 3090 GPU (default: 500). Other hyperparameters include embedding dimension: 128, generator dimensions: (256, 256), discriminator dimensions: (256, 256), epochs: 300, discriminator steps: 1, generator learning rate: 2e-4, and discriminator learning rate: 2e-4. Hyperparameter search tested discriminator steps ([1, 3]), generator learning rate ([1e-4, 2e-4, 4e-4]), and discriminator learning rate ([1e-4, 2e-4, 4e-4]), but defaults performed best. Training took less than 10 hours on a single NVIDIA RTX 3090 GPU without early stopping.

- **ClavaDDPM**: [36] A diffusion-based model for multi-table data generation, Cluster Latent Variable guided Denoising Diffusion Probabilistic Models (ClavaDDPM) leverage Denoising Diffusion Probabilistic Models (DDPMs) to model complex tabular data distributions. By using clustering labels as intermediaries, ClavaDDPM captures long-range dependencies across interconnected tables, particularly through foreign key constraints.

  - **Training and generation**: We adopted the default hyperparameters from the official ClavaDDPM GitHub, featuring a diffusion model with six MLP layers ([512, 1024, 1024, 1024, 1024, 512]), a learning rate of 6e-4, and a batch size of 4096 for 100,000 iterations with a cosine scheduler. The model employs 2000 timesteps with MSE loss and uses 50 clusters for clustering. Training was completed in under 3 hours on a single NVIDIA A6000 GPU.

**Algorithm 1:** Postprocessing for Relational Table Construction

---

**Input:** Set of patients $\mathcal{P}$, Text-based events $E$, timestamps $\mathcal{T}$, valid column names $\mathcal{C}$, valid table names $\mathcal{E}$, observation window $\mathcal{T}_{\max}$
**Output:** Relational tables $\mathcal{R}$
Initialize $\mathcal{R} \leftarrow$ dict of $\tau \rightarrow$ list (a dictionary mapping table names to lists of event records)
Initialize $\mathcal{P}_{\text{valid}} \leftarrow \emptyset$
**for** *each patient $p \in \mathcal{P}$* **do**
    Initialize $\mathcal{R}' \leftarrow$ dict of $\tau \rightarrow$ list (temporary tables for patient $p$)
    Initialize valid_event_count $\leftarrow 0$
    **for** *each event $e_i^p \in E^p$ with timestamp $t_i^p \in \mathcal{T}^p$* **do**
        Parse $e_i^p$ into table name $\tau$ and column-value pairs $\{(c_{i,j}^p, v_{i,j}^p)\}$
        **if** $\tau \notin \mathcal{E}$ **then**
            Continue
        **end**
        **if** *$e_i^p$ does not conform to column-value pair format* **then**
            Continue
        **end**
        **for** *each $(c_{i,j}^p, v_{i,j^p})$ in $\{(c_{i,j}^p, v_{i,j}^p)\}$* **do**
            **if** $c_{i,j}^p \notin \mathcal{C}$ **then**
                $c_{i,j}^p \leftarrow \arg\min_{c \in \mathcal{C}} \text{Levenshtein}(c_{i,j}^p, c)$
            **end**
            **if** *$c_{i,j}^p$ is numeric column and $v_{i,j}^p$ contains non-numeric characters* **then**
                $v_{i,j}^p \leftarrow \text{RemoveNonNumeric}(v_{i,j}^p)$
                **if** *$v_{i,j}^p$ is not numeric* **then**
                    Continue
                **end**
            **end**
        **end**
        Add $(t_i^p, \{(c_{i,j}^p, v_{i,j}^p)\})$ to $\mathcal{R}'[\tau]$
        valid_event_count $\leftarrow$ valid_event_count $+ 1$
    **end**
    **if** *valid_event_count == len($E^p$)* **then**
        Add $p$ to $\mathcal{P}_{\text{valid}}$
        Add $\mathcal{R}'[\tau]$ to $\mathcal{R}[\tau]$
    **end**
**end**
Sort $\mathcal{R}$ by timestamp in ascending order
**for** *each patient $p$ in $\mathcal{P}_{valid}$* **do**
    **for** *each event $(t_i^p, \{(c_{i,j}^p, v_{i,j}^p)\})$ in $\mathcal{R}$* **do**
        **if** *$i > 1$ and $t_i^p < t_{i-1}^p$ or $t_i^p > \mathcal{T}_{max}$* **then**
            Discard $(t_i^p, \{(c_{i,j}^p, v_{i,j}^p)\})$ and all events $(t_k^p, \{(c_{k,j}^p, v_{k,j}^p)\})$ for $k \geq i$ from $\mathcal{R}$
            Break
        **end**
    **end**
**end**
**return** $\mathcal{R}$

---

## E.2 Predictive models for evaluation

We use the following predictive models to assess the quality of synthetic data:

### E.2.1 Clinical utility

- **GenHPF** [15]: GenHPF addresses the heterogeneity of multi-table EHR data by transforming all patient events into a single hierarchical textual sequence. This method sequentially lists events such as medications, infusions, and lab results in chronological order, requiring

Table 10: **Summary of 11 ICU prediction tasks** using synthetic EHR data. Multi-class tasks predict binned lab values, while binary tasks classify whether specific medications are prescribed. Predictions are made at the midpoint ($T/2$) of an observation window of length $T$ (e.g., $T = 12$ hours) to forecast the last lab value or medication event occurrence in the second half.

| Task Name | Source | Task Type |
|---|---|---|
| Creatinine (Cr) | Lab | Multi-class |
| Platelets (Plt) | Lab | Multi-class |
| White Blood Cells (WBC) | Lab | Multi-class |
| Hemoglobin (Hb) | Lab | Multi-class |
| Bicarbonate (HCO3) | Lab | Multi-class |
| Sodium (Na) | Lab | Multi-class |
| Magnesium Sulfate (MgSO4) | Medication | Binary |
| Heparin (Hep) | Medication | Binary |
| Potassium Chloride (KCl) | Medication | Binary |
| Norepinephrine (NE) | Input | Binary |
| Propofol (Prop) | Input | Binary |

minimal preprocessing. GenHPF enables predictive models to capture the full complexity of patient records, making it suitable for multi-task learning without extensive domain-specific feature engineering. Its flexibility supports applications across diverse EHR systems, enhancing model generalizability for downstream predictive tasks.

- **Training**: GenHPF was trained with an embedding dimension of 128, 4 attention heads, and 2 transformer layers. It uses a batch size of 64, a dropout rate of 0.1, and a learning rate of 5e-5 for 50 epochs. Training occurred on a single NVIDIA A6000 GPU, targeting multi-task prediction for EHR lab tests and medications, with early stopping after 10 epochs of no improvement.

- **MEDS-TAB** [38]: MEDS-TAB is an automated tabularization tool that converts Medical Event Data Standard (MEDS)-formatted EHR data into standardized tabular representations by aggregating events into fixed time intervals. It processes irregularly sampled time-series data, supporting various aggregation functions (e.g., sum, count, mean) and window sizes to handle diverse data types, including static codes, numerical values, and time-series events. With high scalability and efficiency—demonstrated by processing 500 patients from MIMIC-IV in 16 seconds with 1,410MB memory usage—MEDS-TAB simplifies predictive modeling by providing structured inputs compatible with machine learning algorithms like XGBoost, achieving competitive performance in clinical tasks.

  - **Training**: MEDS-TAB was trained using the default hyperparameters from the official MEDS-TAB GitHub repository, employing XGBoost for predictive modeling. In contrast to GenHPF, it trains on individual tasks rather than multi-task learning.

### E.2.2 Predictive similarity

For predictive similarity evaluation, the XGBoost model was configured with 100 estimators, a subsample ratio of 0.9, and a maximum bin size of 256. Depending on the task, objectives were set to binary:logistic for binary classification, multi:softmax for multiclass classification, or reg:squarederror for regression, with evaluation metrics logloss, mlogloss, or rmse, respectively.

### E.2.3 Next event prediction

For the Next Event Prediction task, we employ a single-layer LSTM with 128 hidden units, with input and output sizes equal to the number of classes. A sigmoid activation function is used for multi-label classification. The model is trained with the Adam optimizer at a learning rate of 0.001, employing a weighted binary cross-entropy loss (with inverse log-frequency weights normalized to a mean of 1), a batch size of 256, and 50 epochs. A classification threshold of 0.1 is applied to enhance recall for imbalanced classes.

Table 11: **Column-wise fidelity for MIMIC-IV**. Columns are prefixed with "L" (laboratory measurements), "I" (infusions), or "M" (medications). Types "C" and "N" denote categorical and numerical columns, respectively. TRTR (Train-on-Real-Test-on-Real) and TSTR (Train-on-Synthetic-Test-on-Real) denote error rates for categorical columns and SMAPE for numerical columns, reported as mean ± standard deviation (std) across three random seeds, assessing predictive similarity task performance.

| Col. Name | Type | CDE ↓ | I-CDE ↓ | Real Null (%) | Syn Null (%) | TRTR | TSTR |
|---|---|---|---|---|---|---|---|
| L/itemid | C | 0.06 | - | 0.00 | 0.00 | 66.08 ±24.72 | 78.87 ±23.05 |
| L/value | C | 0.07 | 0.14 | 5.78 | 4.51 | 0.00 ±0.00 | 0.77 ±0.02 |
| L/valueuom | C | 0.04 | 0.03 | 13.78 | 12.49 | 0.07 ±0.00 | 2.50 ±0.02 |
| L/flag | C | 0.00 | 0.00 | 62.27 | 64.18 | - | - |
| L/priority | C | 0.03 | 0.04 | 27.34 | 29.48 | 44.16 ±0.08 | 44.32 ±0.19 |
| L/comments | C | 0.11 | 0.22 | 83.41 | 86.15 | 57.34 ±0.61 | 63.52 ±2.04 |
| L/valuenum | N | 0.01 | 0.06 | 11.13 | 9.46 | 116.64 ±0.76 | 86.37 ±1.99 |
| L/ref_range_lower | N | 0.02 | 0.01 | 19.04 | 17.15 | 60.86 ±0.21 | 63.10 ±0.20 |
| L/ref_range_upper | N | 0.01 | 0.01 | 19.04 | 17.15 | 30.45 ±0.66 | 30.01 ±0.36 |
| I/itemid | C | 0.09 | - | 0.00 | 0.00 | 28.17 ±3.03 | 29.19 ±0.13 |
| I/amountuom | C | 0.02 | 0.02 | 0.00 | 0.00 | 0.07 ±0.00 | 0.13 ±0.00 |
| I/rateuom | C | 0.05 | 0.05 | 37.49 | 39.54 | 0.02 ±0.00 | 0.07 ±0.00 |
| I/ordercategoryname | C | 0.03 | 0.03 | 0.00 | 0.00 | 0.16 ±0.00 | 0.19 ±0.00 |
| I/secondaryordercategoryname | C | 0.02 | 0.01 | 25.82 | 26.57 | 0.00 ±0.00 | 0.00 ±0.00 |
| I/ordercomponenttypedescription | C | 0.01 | 0.01 | 0.00 | 0.00 | 0.00 ±0.00 | 0.01 ±0.00 |
| I/ordercategorydescription | C | 0.02 | 0.03 | 0.00 | 0.00 | 0.00 ±0.00 | 0.00 ±0.00 |
| I/totalamountuom | C | 0.00 | 0.00 | 12.35 | 13.59 | - | - |
| I/isopenbag | C | 0.01 | 0.01 | 0.00 | 0.00 | 0.00 ±0.00 | 0.00 ±0.00 |
| I/amount | N | 0.04 | 0.08 | 0.00 | 0.05 | 88.26 ±0.87 | 87.75 ±2.87 |
| I/rate | N | 0.04 | 0.13 | 37.49 | 39.54 | 36.47 ±1.69 | 62.34 ±1.11 |
| I/patientweight | N | 0.09 | - | 0.00 | 0.00 | 21.23 ±0.01 | 21.78 ±0.02 |
| I/totalamount | N | 0.02 | 0.07 | 12.38 | 13.63 | 6.77 ±0.08 | 8.81 ±0.02 |
| I/originalamount | N | 0.03 | 0.12 | 0.17 | 0.10 | 28.00 ±1.23 | 30.78 ±0.34 |
| I/originalrate | N | 0.03 | 0.09 | 5.83 | 4.46 | 41.72 ±3.88 | 48.13 ±2.02 |
| M/drug | C | 0.08 | - | 0.00 | 0.00 | 31.00 ±0.38 | 40.53 ±1.01 |
| M/drug_type | C | 0.00 | 0.00 | 0.00 | 0.00 | 0.82 ±0.07 | 1.41 ±0.04 |
| M/prod_strength | C | 0.08 | 0.05 | 0.09 | 0.04 | 17.69 ±0.63 | 37.79 ±0.43 |
| M/dose_unit_rx | C | 0.03 | 0.02 | 0.09 | 0.04 | 5.93 ±0.11 | 13.06 ±0.28 |
| M/form_unit_disp | C | 0.03 | 0.04 | 0.09 | 0.05 | 8.16 ±0.21 | 21.07 ±0.45 |
| M/route | C | 0.04 | 0.04 | 0.09 | 0.04 | 31.99 ±0.49 | 40.59 ±0.30 |
| M/doses_per_24_hrs | N | 0.02 | 0.06 | 59.97 | 61.97 | 60.30 ±0.11 | 60.36 ±0.12 |
| M/dose_val_rx | N | 0.02 | 0.04 | 6.13 | 6.38 | 109.53 ±0.20 | 105.76 ±0.57 |
| M/form_val_disp | N | 0.02 | 0.03 | 5.59 | 5.99 | 129.93 ±13.27 | 82.49 ±4.43 |

# F    Additional experimental results

## F.1    Column-wise fidelity metrics

We report column-wise fidelity metrics for `RawMed`-generated data on the MIMIC-IV and eICU datasets, evaluating distributional similarity and predictive performance across categorical and numerical columns. Notably, `RawMed`-generated null ratios closely align with real data, as detailed in Tables 11 and 12.

## F.2    Task-specific predictive performance

We report task-wise predictive performance for clinical tasks using `RawMed`-generated data on the MIMIC-IV and eICU datasets, comparing Train-on-Real-Test-on-Real (TRTR) and Train-on-Synthetic-Test-on-Real (TSTR) metrics, as detailed in Table 13.

## F.3    ED dataset generalizability

To assess the generalizability of `RawMed` across diverse clinical environments, we evaluated its performance on the Emergency Department (ED) dataset from MIMIC-IV-ED, applying the same methodology and evaluation framework as used for the ICU dataset.

Table 12: **Column-wise fidelity for eICU**. Columns are prefixed with "L" (laboratory measurements), "I" (infusions), or "M" (medications). Types "C" and "N" denote categorical and numerical columns, respectively. TRTR (Train-on-Real-Test-on-Real) and TSTR (Train-on-Synthetic-Test-on-Real) denote error rates for categorical columns and SMAPE for numerical columns, reported as mean ± standard deviation (std) across three random seeds, assessing predictive similarity task performance.

| Col. Name | Type | CDE ↓ | I-CDE ↓ | Real Null (%) | Syn Null (%) | TRTR | TSTR |
|---|---|---|---|---|---|---|---|
| L/labname | C | 0.04 | - | 0.00 | 0.00 | 52.57 ±3.92 | 53.10 ±12.26 |
| L/labmeasurenamesystem | C | 0.03 | 0.07 | 5.24 | 4.82 | 0.07 ±0.02 | 0.59 ±0.07 |
| L/labmeasurenameinterface | C | 0.04 | 0.14 | 6.37 | 5.59 | 54.97 ±6.30 | 49.94 ±0.49 |
| L/labtypeid | C | 0.03 | 0.06 | 0.00 | 0.00 | 0.00 ±0.00 | 0.28 ±0.04 |
| L/labresult | N | 0.01 | 0.11 | 0.93 | 0.64 | 14.68 ±0.55 | 24.54 ±0.65 |
| L/labresulttext | N | 0.01 | 0.12 | 0.75 | 0.54 | 19.72 ±0.46 | 23.40 ±0.35 |
| I/drugname | C | 0.16 | - | 0.00 | 0.00 | 57.38 ±0.32 | 60.48 ±0.20 |
| I/infusionrate | N | 0.03 | 0.43 | 55.54 | 63.14 | 77.70 ±0.27 | 77.95 ±1.08 |
| I/drugamount | N | 0.06 | 0.32 | 67.48 | 74.43 | 71.46 ±1.57 | 69.29 ±0.18 |
| I/volumeoffluid | N | 0.03 | 0.23 | 67.47 | 74.37 | 14.80 ±0.47 | 18.11 ±0.18 |
| I/patientweight | N | 0.03 | - | 90.91 | 95.91 | 12.49 ±0.20 | 33.41 ±0.44 |
| I/drugrate | N | 0.03 | 0.36 | 0.46 | 0.55 | 93.12 ±0.66 | 92.25 ±0.25 |
| M/drugname | C | 0.10 | - | 32.53 | 29.65 | 73.75 ±0.17 | 81.10 ±0.47 |
| M/drugivadmixture | C | 0.02 | 0.03 | 0.00 | 0.00 | 14.49 ±0.07 | 16.36 ±0.01 |
| M/dosage | C | 0.07 | 0.15 | 11.13 | 11.38 | 80.19 ±0.08 | 81.74 ±0.17 |
| M/routeadmin | C | 0.07 | 0.11 | 0.00 | 0.01 | 55.23 ±0.15 | 57.59 ±0.09 |
| M/frequency | C | 0.08 | 0.22 | 12.60 | 11.31 | 85.52 ±0.19 | 88.22 ±0.10 |
| M/prn | C | 0.00 | 0.06 | 0.01 | 0.08 | 10.40 ±0.12 | 11.97 ±0.10 |

Table 13: **Clinical task utility evaluation**. Train-on-Synthetic-Test-on-Real (TSTR) performance for MIMIC-IV and eICU datasets, reported as mean ± standard deviation (std) across three random seeds. TRTR denotes Train-on-Real-Test-on-Real, and TSTR denotes Train-on-Synthetic-Test-on-Real. GenHPF and MEDS-TAB are synthetic data generation methods. Tasks include laboratory value predictions (Creatinine: Cr, Platelets: Plt, White Blood Cells: WBC, Hemoglobin: Hb, Bicarbonate: HCO3, Sodium: Na) and medication/input predictions (Magnesium Sulfate: MgSO4, Heparin: Hep, Potassium Chloride: KCl, Norepinephrine: NE, Propofol: Prop).

| | MIMIC-IV | | | | eICU | | | |
|---|---|---|---|---|---|---|---|---|
| | GenHPF | | MEDS-TAB | | GenHPF | | MEDS-TAB | |
| Task | TRTR | TSTR | TRTR | TSTR | TRTR | TSTR | TRTR | TSTR |
| Cr | 0.89 ±0.00 | 0.88 ±0.00 | 0.97 ±0.00 | 0.95 ±0.00 | 0.86 ±0.00 | 0.83 ±0.00 | 0.94 ±0.00 | 0.89 ±0.00 |
| Plt | 0.88 ±0.01 | 0.87 ±0.00 | 0.96 ±0.00 | 0.95 ±0.00 | 0.84 ±0.00 | 0.83 ±0.00 | 0.94 ±0.00 | 0.91 ±0.00 |
| WBC | 0.78 ±0.01 | 0.73 ±0.03 | 0.91 ±0.00 | 0.86 ±0.00 | 0.73 ±0.01 | 0.70 ±0.02 | 0.87 ±0.00 | 0.80 ±0.00 |
| HB | 0.75 ±0.01 | 0.74 ±0.01 | 0.85 ±0.00 | 0.82 ±0.00 | 0.74 ±0.00 | 0.72 ±0.01 | 0.81 ±0.00 | 0.78 ±0.00 |
| HCO3 | 0.79 ±0.01 | 0.79 ±0.01 | 0.90 ±0.00 | 0.90 ±0.00 | 0.78 ±0.01 | 0.76 ±0.00 | 0.88 ±0.00 | 0.86 ±0.00 |
| Na | 0.80 ±0.01 | 0.78 ±0.01 | 0.93 ±0.00 | 0.92 ±0.00 | 0.80 ±0.01 | 0.79 ±0.01 | 0.91 ±0.00 | 0.90 ±0.00 |
| MgSO4 | 0.81 ±0.02 | 0.81 ±0.01 | 0.86 ±0.00 | 0.83 ±0.00 | 0.73 ±0.01 | 0.72 ±0.01 | 0.81 ±0.01 | 0.75 ±0.00 |
| Hep | 0.68 ±0.01 | 0.65 ±0.00 | 0.80 ±0.00 | 0.74 ±0.00 | 0.65 ±0.00 | 0.63 ±0.01 | 0.72 ±0.00 | 0.66 ±0.00 |
| KCl | 0.73 ±0.01 | 0.70 ±0.00 | 0.80 ±0.00 | 0.73 ±0.00 | 0.74 ±0.01 | 0.66 ±0.00 | 0.77 ±0.00 | 0.68 ±0.00 |
| NE | 0.96 ±0.00 | 0.96 ±0.00 | 0.96 ±0.00 | 0.96 ±0.00 | 0.96 ±0.00 | 0.95 ±0.00 | 0.95 ±0.00 | 0.94 ±0.00 |
| Prop | 0.94 ±0.00 | 0.94 ±0.00 | 0.95 ±0.00 | 0.94 ±0.00 | 0.96 ±0.00 | 0.95 ±0.00 | 0.96 ±0.00 | 0.94 ±0.00 |
| **Avg** | **0.82** ±0.09 | **0.80** ±0.09 | **0.90** ±0.06 | **0.87** ±0.08 | **0.80** ±0.09 | **0.78** ±0.10 | **0.87** ±0.08 | **0.83** ±0.10 |

**Predictive task definition** Due to differences in clinical context, the ICU task definitions were not directly applicable to the ED dataset. Instead, we designed a binary classification task to evaluate the utility of synthetic data, based on four clinically relevant features: heart rate, respiratory rate, morphine administration, and ondansetron administration. The observation window was set to $T = 6$ hours, split into two 3-hour segments. The first 3 hours serve as input for the predictive model, with predictions made at the midpoint ($T/2 = 3$ hours). Binary labels were defined as follows, using the maximum value within the second 3-hour period for heart rate and respiratory rate, and occurrence for medication administration:

Table 14: Comparison of VQ-VAE and RQ-VAE on MIMIC-IV and eICU, reporting Column-wise Density Estimation (CDE), Pairwise Column Correlation (PCC), Time Gap, and Event Count metrics.

| | MIMIC-IV | | eICU | |
|---|---|---|---|---|
| Metric | VQ | RQ | VQ | RQ |
| CDE ↓ | 0.05 | 0.04 | 0.04 | 0.05 |
| - Categorical | 0.05 | 0.04 | 0.05 | 0.06 |
| - Numerical | 0.05 | 0.03 | 0.03 | 0.03 |
| - Patientweight | 0.28 | 0.09 | 0.05 | 0.03 |
| PCC ↓ | 0.04 | 0.04 | 0.07 | 0.06 |
| Time Gap ↓ | 0.05 | 0.01 | 0.03 | 0.03 |
| # Events ↓ | 0.13 | 0.02 | 0.03 | 0.05 |

Table 15: Scalability results on MIMIC-IV and eICU for 6-hour and 24-hour observation windows, reporting Column-wise Density Estimation (CDE), Pairwise Column Correlation (PCC), Time Gap, and Event Count metrics.

| | MIMIC-IV | | eICU | |
|---|---|---|---|---|
| Obs. size | 6h | 24h | 6h | 24h |
| CDE ↓ | 0.04 | 0.05 | 0.06 | 0.06 |
| PCC ↓ | 0.03 | 0.02 | 0.09 | 0.07 |
| Time Gap ↓ | 0.05 | 0.07 | 0.01 | 0.02 |
| # Events ↓ | 0.07 | 0.11 | 0.08 | 0.01 |

- **Heart rate** (beats per minute, bpm): 1 if the maximum value is $\geq$ 120 bpm (indicating tachycardia, above the normal adult range of 60–100 bpm), 0 otherwise.

- **Respiratory rate** (respirations per minute, rpm): 1 if the maximum value is $>$ 24 rpm (indicating tachypnea, beyond the normal adult range of 12–20 rpm), 0 otherwise.

- **Morphine administration**: 1 if administered, 0 if not.

- **Ondansetron administration**: 1 if administered, 0 if not.

These thresholds were selected based on established clinical criteria to identify critical conditions in the ED setting.

**Results**   The evaluation demonstrated that `RawMed` generalizes effectively to the ED dataset, achieving robust performance across predictive tasks. Table 16 compares `RawMed` with baseline models (*i.e.*, SDV, RC-TGAN, ClavaDDPM), reporting fidelity metrics (Column-wise Density Estimation, CDE; Pairwise Column Correlation, PCC; Time Gap; Number of Events) and clinical utility. `RawMed` surpasses baselines in most metrics, with a CDE of 0.08, PCC of 0.02, Time Gap of 0.02, Number of Events error of 0.02, and a Utility score of 0.79, nearing the real data's Utility of 0.83. Despite differing clinical environments, `RawMed`'s performance closely matches real data utility and surpasses baselines, indicating potential for generalization to other datasets.

## F.4   Conditional generation

We conducted experiments on conditional generation by incorporating static features into the Temporal Modeling framework (Section 3.3). Specifically, we integrated age, gender, and admission diagnosis (or type) as static features. During training, these features were prepended to the sequences, enabling the model to learn the combined sequence auto-regressively. For generation, the model utilized real static features as conditions to produce the complete sequence. To further investigate conditional generation, we explored the inclusion of initial medical context, providing the model with real static features and approximately one-fourth of the initial real-data events from the sequence, tasking it with generating the remaining three-fourths.

Table 16: Model Performance Comparison on MIMIC-IV-ED, reporting Column-wise Density Estimation (CDE), Pairwise Column Correlation (PCC), Time Gap, Event Count, and Clinical Utility (AUROC) metrics. The best results are in bold.

| Model | CDE ↓ | PCC ↓ | Time Gap ↓ | # Events ↓ | Utility ↑ |
|---|---|---|---|---|---|
| Real | – | – | – | – | 0.83±0.00 |
| SDV | **0.05** | 0.21 | 0.46 | 0.07 | 0.63±0.09 |
| RC-TGAN | 0.55 | 0.20 | 0.34 | 0.06 | 0.57±0.06 |
| ClavaDDPM | 0.16 | 0.11 | 0.35 | 0.06 | 0.64±0.06 |
| RawMed | 0.08 | **0.02** | **0.02** | **0.02** | **0.79**±0.05 |

To assess whether the generated sequences effectively captured the relationships with static features, we trained three classifiers to predict each age, gender, admission diagnosis or type from the generated sequences, excluding the static feature portion itself. For this classification, age was binned, and for the admission diagnosis, only four specific categories from the 'Non-operative Organ Systems' hierarchy (which account for the majority of diagnoses) were selected, namely cardiovascular, neurologic, respiratory, and gastrointestinal.

**Results**  The performance of our conditional generation approach on the MIMIC-IV and eICU datasets is presented in Table 17. For the MIMIC-IV dataset, the synthetic data demonstrates performance comparable to real data, indicating that our model effectively captures the relationship between static features and sequential data. Notably, conditioning on initial events enhances performance, with the AUROC for age prediction increasing from 0.75 to 0.77. For gender prediction, both real and synthetic data achieved an AUROC of 1.00. This is attributable to the inclusion of all columns from the *labevents* table in our synthetic data generation, which unintentionally incorporates gender-specific reference ranges in lab values (e.g., creatinine, hemoglobin) within the *ref_range_lower* and *ref_range_upper* columns, serving as direct gender indicators.

On the eICU dataset, similar trends were observed, with synthetic data approaching the performance of real data, particularly when leveraging initial medical context. The AUROC values improved when conditioning on initial events, rising from 0.65 to 0.69 for age, 0.52 to 0.59 for gender, and 0.86 to 0.88 for admission diagnosis, compared to static features alone. These results underscore the flexibility of our framework in supporting conditional generation with richer feature sets, enhancing synthetic data quality. Although gender prediction on eICU exhibited a slight performance drop compared to real data, we anticipate improvements through further methodological refinements and hyperparameter optimization. The results from both datasets validates the flexibility and potential of our approach for conditional generation.

Table 17: Performance (AUROC) of conditional generation models on MIMIC-IV and eICU datasets. Adm. Type refers to Admission Type, and Adm. Diag. refers to Admission Diagnosis.

| Model | MIMIC-IV | | | eICU | | |
|---|---|---|---|---|---|---|
| | Age | Gender | Adm. Type | Age | Gender | Adm. Diag. |
| Real | 0.80 | 1.00 | 0.92 | 0.72 | 0.67 | 0.89 |
| RawMed (static only) | 0.75 | 1.00 | 0.89 | 0.65 | 0.52 | 0.86 |
| RawMed (static+1/4 initial events) | 0.77 | 1.00 | 0.89 | 0.69 | 0.59 | 0.88 |

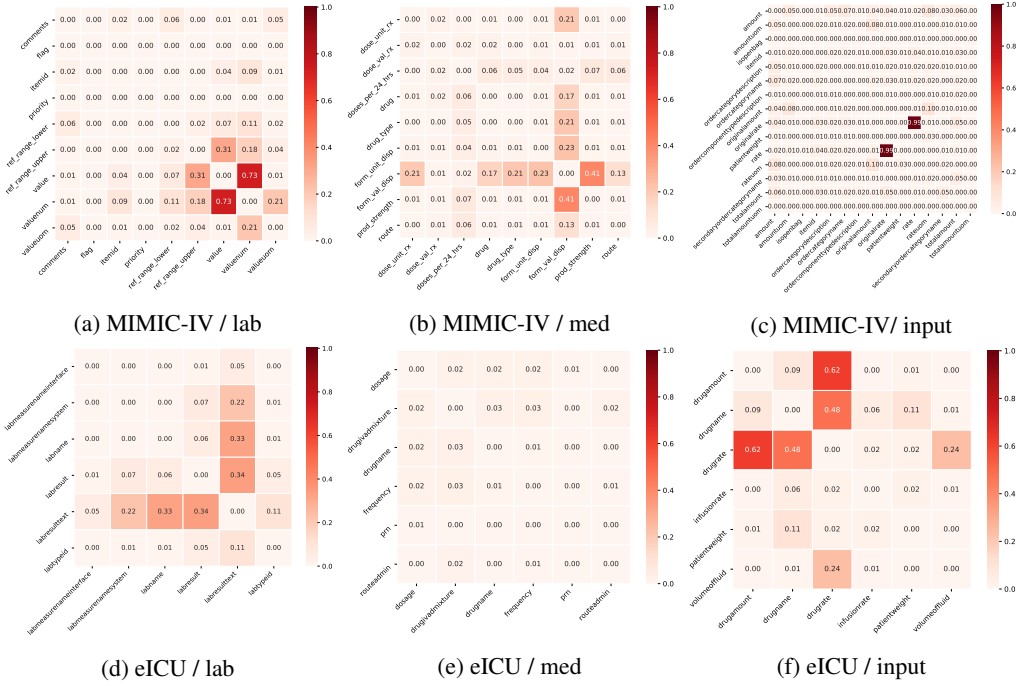

Figure 3: Table-specific visualizations of absolute differences in Pairwise Column Correlation (PCC) matrices between real and synthetic data for MIMIC-IV and eICU tables.

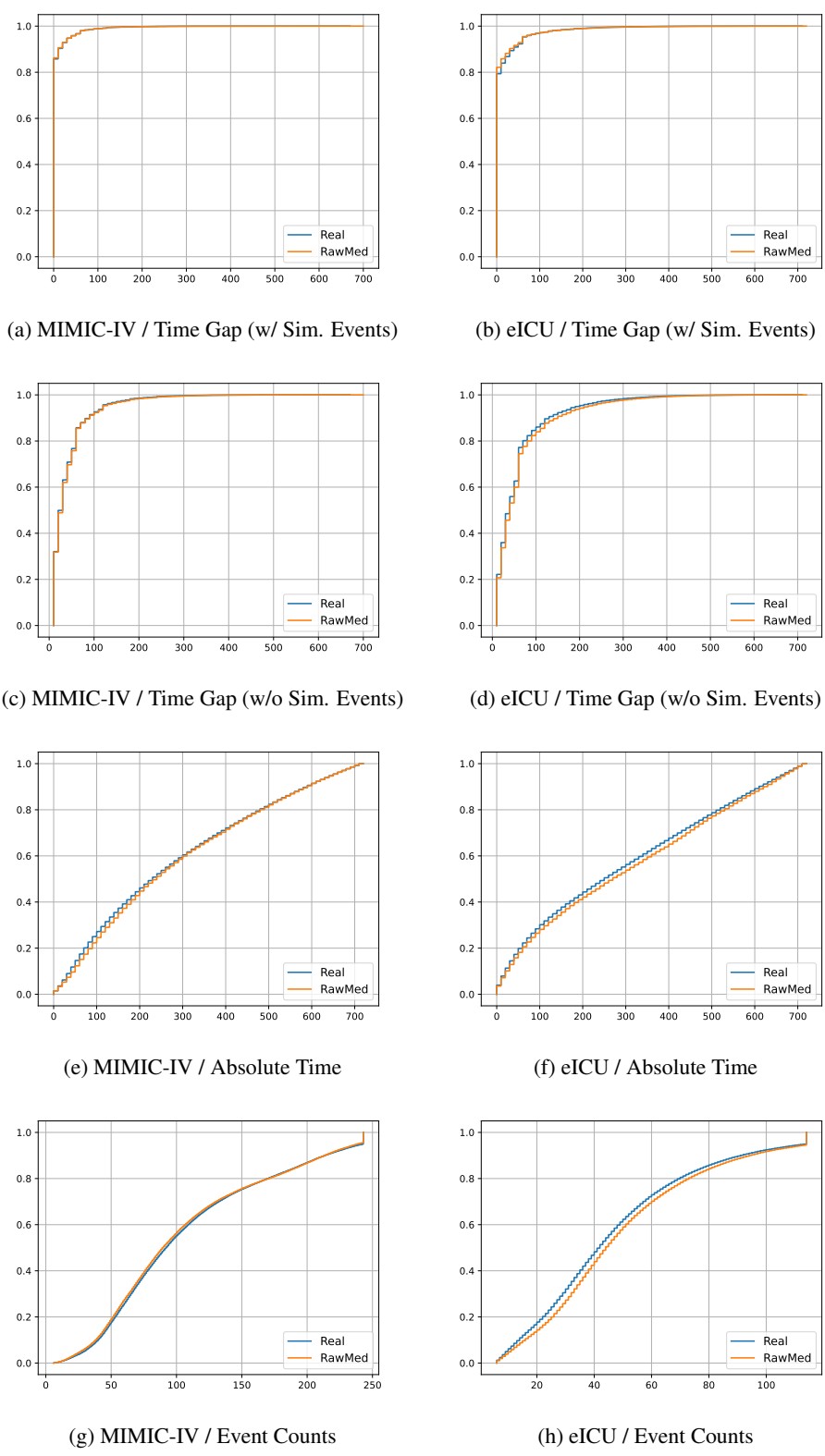

(a) MIMIC-IV / Time Gap (w/ Sim. Events)   (b) eICU / Time Gap (w/ Sim. Events)

(c) MIMIC-IV / Time Gap (w/o Sim. Events)   (d) eICU / Time Gap (w/o Sim. Events)

(e) MIMIC-IV / Absolute Time   (f) eICU / Absolute Time

(g) MIMIC-IV / Event Counts   (h) eICU / Event Counts

Figure 4: Cumulative Distribution Functions (CDFs) of real and synthetic data for MIMIC-IV and eICU datasets, showing Time Gap (with and without simultaneous events), Absolute Time from admission, and Event Counts per patient.

