# OpenReview forum: "Generating Multi-Table Time Series EHR from Latent Space with Minimal Preprocessing"
_NeurIPS.cc/2025/Conference — NeurIPS 2025 poster_

### Official Review · Reviewer_Uzbn · 2025-06-28

**Clarity:** 3
**Significance:** 2
**Originality:** 3
**Rating:** 4
**Confidence:** 4

**Summary:**

This paper proposes RawMed, which recasts time-series electronic health records (EHRs) as a sequence of events, trains a Transformer to auto-regressively predict the next event in a sequence, and uses the pre-trained transformer to generate synthetic time-series EHRs. Although autoregressive transformers have previously been applied to generating tabular data and synthetic time-series EHRs at the temporal resolution of visits, this is the first time they have been applied to generating time-series EHRs at the original temporal resolution of the data. The raw data is prepared for the Transformer using a textual embedding, BERT-based tokenizer, and pre-trained encoder. After applying the reverse process to the output of the Transformer, the generated sequence of events is cleaned to ensure it corresponds to a feasible EHR, and finally the generated sequence of events is converted into a synthetic EHR.

RawMed is evaluated using a dataset comprising three time-series tables from the MIMIC-IV and eICU datasets. Due to the lack of an existing benchmark for generating multi-table, time-series EHRs, generic multi-table generation methods such as RC-TGAN and ClavaDDPM are used as baselines, as well as RealTabFormer, a fine-tuned large language model. RawMed outperforms the baseline methods across a range of metrics designed to test the distributional similarity of the generated data and it's effectiveness as training data for downstream tasks.

**Questions:**

## Questions

- What percentage of generated events were modified or rejected and what percentage of generated patients were rejected by the post-processing steps?

- Where applicable, were the post-processing steps applied to the baseline models?

- How was the $k$ in the top$-k$ sampling chosen? It feels like the distribution of the generated data depends hugely on this choice, but the impact of varying $k$ is never analysed.

## Minor Comments

- You should include at least a brief mention of HALO (see [here](https://www.nature.com/articles/s41467-023-41093-0.pdf)), which uses a Transformer architecture to generate synthetic EHRs with visit level temporal resolution.

- On line 87 you use a double negative.

**Ethical Concerns:**

["NO or VERY MINOR ethics concerns only"]

**Final Justification:**

The authors have addressed my concerns regarding the separation between lossy and lossless preprocessing. Overall, I feel the paper makes a significant contribution to one sub-area of machine learning. I am therefore maintaining my score, and recommending the paper for acceptance.

**Limitations:**

Yes.

**Paper Formatting Concerns:**

None.

**Quality:**

3

**Strengths And Weaknesses:**

## Strengths

- The paper proposes a unique approach for generating synthetic EHRs, the chosen datasets and baselines are suitable, and the set of evaluation metrics is extensive, although not necessarily complete (see weaknesses).
- The proposed method achieves strong performance across the whole range of evaluation metrics.
- The paper is well written.

## Weaknesses

- The paper repeatedly highlights the 'minimal preprocessing' of the data as a benefit of RawMed. I have two issues with this:
  1. The authors never make it clear why preprocessing is a problem. In particular, they include one-hot encoding and normalisation as examples of the heavy preprocessing other methods require (lines 40 and 89), but these are lossless transformations. I believe the authors wish to make the point that preprocessing steps such as numerical binning and aggregation lose information and RawMed avoids operations like these. However,
  2. RawMed uses extensive preprocessing to prepare the data for the Transformer, including a textual embedding, BERT-based tokenizer, and pre-trained encoder, the last of which loses information, as can be seen in the change of the patientweight distribution after reconstruction in Figure 2. Furthermore, the paper does not evaluate the accuracy of the reconstruction.

- A possible disadvantage to using an auto-regressive next token predictor to generate synthetic EHRs is the model under-representing low probability events. The proposed suite of evaluation metrics does not test the generated data for this failure point.

---

> ### Author Rebuttal · Authors · 2025-07-31
>
> **Response to W1-1:**
> Thanks for the opportunity to clarify our "minimal preprocessing" claim. Our intent is to highlight how RawMed significantly reduces reliance on lossy preprocessing steps that are prevalent in existing EHR synthesis methods.
>
> To clarify what constitutes this "lossy preprocessing", we refer to operations that fundamentally distort or reduce the original data. This includes techniques like binning continuous values or aggregating data into fixed time intervals, which sacrifice the original data's granularity. Similarly, term normalization can eliminate subtle distinctions present in the raw information by grouping or standardizing terms. We agree that one-hot encoding is a lossless transformation and will clarify this distinction in our paper.
>
> Crucially, the first major limitation we pinpoint in existing works is their reliance on feature selection. This is a prime example of extensive, lossy preprocessing. It involves domain experts manually choosing only a subset of features they deem relevant for specific downstream tasks, thereby discarding vast amounts of original, potentially valuable information from the outset.
>
> The core issue with all these lossy transformations—be it feature selection, binning, aggregation, or certain normalizations—is the inevitable reduction in data utility. For instance, data aggregated to hourly "averages" cannot accurately answer queries requiring precise "counts" within that hour, nor can it support analyses demanding finer temporal granularity. When the diverse future uses of synthetic data are unknown, losing original information inherently limits its overall versatility and applicability. RawMed directly addresses this by avoiding such transformations, thus preserving the richness of the raw data for broader utility.
>
> ---
>
> **Response to W1-2:**
> **RawMed's preprocessing is not extensive**. Unlike other methods requiring complex, domain-specific feature engineering, RawMed unifies all data into a text format. It then applies standard, general-purpose steps like text embedding and tokenization. This significantly reduces the overall preprocessing burden and complexity, making our approach comparatively minimal.
>
> **Furthermore, RawMed's preprocessing is near-lossless**. Tokenization and text embedding primarily change the data's representation, not its content. Any potential information loss can occur during the compression stage within our generative modeling part, not during the initial preprocessing. The encoder’s ability to preserve information is empirically validated by token-level reconstruction accuracy on the test set, reaching 98.2% on MIMIC-IV and 99.8% on eICU, which indicates minimal information loss.
>
> We hope this response fully addresses your questions regarding our stance on preprocessing. We will also incorporate these clarifications and discussions into the introduction and Section 2.1 (Related Work) of our paper to provide a more comprehensive explanation.
>
> ---
>
> **Response to W2:**
> We appreciate your valid concern regarding the potential for auto-regressive models to under-represent low-probability events, and the lack of a direct metric for this in our current evaluation suite. However, our metrics indirectly address this point. Specifically, our CDE (Column-wise Density Estimation) metric, when applied to item-related columns (e.g., specific lab tests, medications), measures the marginal distribution similarity across all possible events within that column. Additionally, our I-CDE (Item-wise CDE) metric extends this by averaging CDE across various item types, encompassing relatively rare drug classes and lab test varieties. In both CDE and I-CDE, RawMed consistently demonstrated superior fidelity to the real data compared to other baseline models.
>
> ---
>
> **Response to Q1:**
> Our post-processing pipeline involves two main stages: converting text to tabular format and enhancing data quality.
>
> In the "converting text to tabular format" stage, we observed the following event-level modification and rejection rates: 0.42% modified and 0.07% rejected in eICU, and 0.73% modified and 0.17% rejected in MIMIC-IV. At the patient level, rejection rates were notably higher: 2.71% in eICU and 18.23% in MIMIC-IV.
>
> For the "enhancing data quality" stage, stricter quality checks led to higher modification and rejection rates at the event level. In eICU, approximately 1.39% of generated events were modified and 0.49% were rejected. For MIMIC-IV, 1.07% of events were modified and 0.79% were rejected. These stringent checks resulted in significantly elevated patient-level rejection rates: 16.90% in eICU and 54.53% in MIMIC-IV.
>
> The significantly higher patient-level rejection rates, compared to event-level rejections across both stages, are a direct result of our stringent data integrity policy. A single erroneous event leads to the rejection of an entire patient's data. For instance, MIMIC-IV patients can have up to 243 events, all of which must be accurate to ensure the patient's data is retained within the dataset.
>
> ---
>
> **Response to Q2:**
> We appreciate you raising this point. We applied an equivalent level of post-processing to the RealTabFormer baseline in Table 5. However, extending this same strict post-processing to all baseline models, particularly those in Table 2, 3, and 4, turned out to be challenging and largely impractical. For example, applying our full post-processing pipeline to baseline models trained on MIMIC-IV resulted in approximately 99% of their generated data being discarded. This high rejection rate indicates that most baseline models couldn't perfectly generate all patient events under our strict post-processing standards.
>
> Nevertheless, to ensure a fair comparison of model performance alone, we applied a less stringent post-processing to RawMed, resulting in an output quality comparable to the baselines in Tables 2, 3, and 4.
>
> The resulting performance is detailed in Tables A, B and C, which can be directly compared with the results in Tables 2, 3, and 4 of our main paper. We could not include those results here due to space limitations.
>
> Generally, the performance for most metrics remains similar to our prior results, where full post-processing was applied to RawMed. However, specifically for MIMIC-IV, our model's PCC and SMAPE values are slightly higher than our prior results and also higher than ClavaDDPM, indicating slightly worse performance in these metrics. Despite this, across all other metrics and consistently compared to other baselines, our model demonstrates superior performance.
>
> However, we emphasize that full post-processing is crucial for generating high-integrity data with RawMed. We do not recommend using partial post-processing, as it fundamentally compromises the quality of the synthetic data.
>
>
> **Table A:  Results of single-table evaluation on MIMIC-IV and eICU datasets.**
>
> | Dataset | CDE   | I-CDE | PCC   | I-PCC | ER    | SMAPE  |
> |---------|------:|------:|------:|------:|------:|-------:|
> | mimiciv |  0.03 |  0.05 |  0.09 |  0.15 | 18.74 | 106.09 |
> | eicu    |  0.05 |  0.12 |  0.07 |  0.13 | 49.44 |  62.12 |
>
> **Table B: Results of clinical utility evaluation on MIMIC-IV and eICU datasets**
>
> | Dataset | MEDS-TAB      | GenHPF      |
> |---------|---------------|-------------|
> | mimiciv | 0.87 ± 0.08 | 0.80 ± 0.10 |
> | eicu    | 0.83 ± 0.09 | 0.78 ± 0.10 |
>
> **Table C: Results of temporal fidelity evaluation on MIMIC-IV and eICU datasets**
>
> | Dataset | Next Event Predict (F1) | Time Gap | Event Count |
> |---------|---------|---------:|------------:|
> | mimiciv | 0.15 ± 0.00 |   0.01 |        0.05 |
> | eicu    |    0.25 ± 0.03 |  0.02 |        0.06 |
>
>
> ---
>
> **Response to Q3:**
> Thank you for your question about how we chose the k-value in top-k sampling and your valuable point about the lack of analysis on its varying impact on generated data. You're absolutely right that the choice of k significantly influences the data's characteristics.
>
> Initially, we conducted preliminary, small-scale experiments using the MIMIC-IV and eICU dataset, drawing inspiration from autoregressive sampling in NLP. We started with small k-values like 5, 10, and 15. However, we realized these values didn't accurately reflect the event distribution of the real data. This led us to intentionally increase k, arriving at our chosen values through a more heuristic approach rather than a full, systematic analysis at the start. While we didn't perform a complete, systematic analysis in the beginning, your insightful question prompted us to look closer at neighboring k-values.
>
> Due to time constraints, we've focused on two immediate surrounding values, but these preliminary results already highlight the impact on performance: For the eICU dataset, a k-value of 150 yielded the highest TSTR AUROC at 0.751, significantly outperforming k=100 (0.728) and k=200 (0.723). Similarly, for the MIMIC-IV dataset, a k-value of 250 demonstrated the best performance with a TSTR AUROC of 0.797, notably better than k=200 (0.773) and k=300 (0.776). We acknowledge the critical role of k and are committed to providing a more comprehensive analysis with a broader grid search during the discussion period, continually updating our findings to further solidify justification for these choices.

---

> > ### Comment · Reviewer_Uzbn · 2025-08-04
> > **Rebuttal Reply**
> >
> > Thank you for your thorough response.
> >
> > ## Pre-processing
> >
> > Given the broad nature of pre-processing, I think an expanded discussion around lossy and loss-less pre-processing, and the differences between pre-processing steps such as feature selection and tokenization/textual embeddings, will greatly benefit the paper. I still find the repeated claim of `minimal preprocessing' hard to justify, given that your model does include significant (loss-less) pre-processing. Additionally, whether the compression is part of the generative model or pre-processing is debatable, given that it is trained separately. Many people wouldn't include pre-trained textual embeddings as part of the model, but these play a similar role. Therefore, I do not believe it is unreasonable to take the view that your full pipeline includes lossy pre-processing as well, although as you show the amount of lost information is small.
> >
> > ## Post-processing and Top-k
> >
> > Thank you for the expanded results. I believe including a full ablation on the impact of the post-processing and the k-value in top-k will significantly improve the paper.
> >
> > ---
> >
> > Overall, I am happy with the suggested changes, and will be continuing to recommend acceptance.

---

> > > ### Author Response · Authors · 2025-08-05
> > > **Response to Reviewer Comments**
> > >
> > > Thank you for your insightful and thorough feedback. We greatly appreciate your detailed comments, which have helped us to improve the clarity of our manuscript. We agree with your points regarding the comprehensive nature of preprocessing and the importance of distinguishing between lossy and lossless preprocessing. We also fully concur with your suggestion to perform an ablation study on post-processing and the top-k value, and we will incorporate this analysis to enhance the completeness of our study.
> > >
> > > We have carefully considered your concerns regarding the term "minimal preprocessing" and whether the compression stage should be considered preprocessing. We would like to clarify our position on these two points below.
> > >
> > > **Clarification of "minimal preprocessing":**
> > > Our intention in using the term "minimal preprocessing" was not to suggest that there is little or no preprocessing. Instead, we used this term to emphasize that our approach uses the raw data as directly as possible, applying transformations that preserve the original data's content and structure.
> > > RawMed intentionally avoids the lossy preprocessing steps common in existing EHR synthesis methods, such as feature selection or binning, which often require domain expertise. Instead, we employ universal, lossless preprocessing steps to preserve the integrity of the original data.
> > >
> > > We acknowledge that the term "minimal preprocessing" can be ambiguous. To address this, we will clarify its meaning in the manuscript by replacing it with a more precise phrase like "minimal lossy preprocessing" or by explicitly defining the term within the context of our work.
> > >
> > > **Compression from a generative modeling perspective:**
> > > Your feedback on the potential interpretation of compression as a preprocessing step was crucial in helping us to re-evaluate the definitions of our pipeline stages. It helped us to clarify our approach, and we'd like to provide a clearer breakdown of the stages.
> > >
> > > - Textualization: This stage converts the raw EHR data into a text sequence without altering its content.
> > >
> > > - Tokenization and Embedding: The text is then converted into a model input format using a lookup table, minimizing information loss.
> > >
> > > - Generative Modeling: In this stage, we use a VQ-VAE to learn and compress the embedded data within a latent space. Any information loss in this stage stems from the modeling design rather than preprocessing, thereby distinguishing compression from preprocessing steps.
> > >
> > > We hope this clarification resolves your concern, and we would greatly appreciate your feedback on whether this delineation adequately addresses the issue. Thank you once again for your invaluable feedback.

---

> > > > ### Comment · Reviewer_Uzbn · 2025-08-05
> > > >
> > > > Thank you for your response.
> > > >
> > > > The additional clarifications provided have settled my concerns; I find the distinction between the pre-processing of RawMed and alternative models clearer now. I believe the including an expanded discussion on pre-processing, outlining the points discussed above, will benefit the paper.

---

> > > > > ### Author Response · Authors · 2025-08-06
> > > > > **Response to Reviewer Comments**
> > > > >
> > > > > Thank you. We will incorporate an expanded discussion on the pre-processing as you've suggested.
> > > > >
> > > > > We appreciate you taking the time to review our rebuttal. Your valuable feedback has helped us clarify this important distinction. Please let us know if there are any remaining concerns, and we will do our best to address them.

---

### Official Review · Reviewer_tPMX · 2025-07-01

**Clarity:** 3
**Significance:** 3
**Originality:** 2
**Rating:** 4
**Confidence:** 3

**Summary:**

This paper introduces RawMed, a framework for generating synthetic electronic health records (EHRs) that preserves the multi-table relational structure and time-series nature of raw EHR data.

Previous approaches that generate only selected features with extensive preprocessing, RawMed represents EHR data as text, employs Residual Quantization for compression, and uses an autoregressive model to capture temporal dynamics.

The authors conduct comprehensive evaluation on MIMIC-IV and eICU dataset, showing that RawMed outperforms baseline methods in fidelity, utility, and privacy protection.

**Questions:**

1. Could the authors provide more insight into the errors that require postprocessing? Understanding the nature of these errors might reveal fundamental limitations of the text-based approach.
2. How might conditional generation be incorporated into the framework? This would significantly enhance the practical utility of the approach. Being said, can this approach extended to hybrid generation mode with user preferences?
3. Information Loss: When compressing raw EHR data using RQ, what is the quantitative information loss measured across different column types? And what does that mean?

**Ethical Concerns:**

["NO or VERY MINOR ethics concerns only"]

**Final Justification:**

Authors addressed some concerns of mine. I will maintain the score

**Limitations:**

Yes

**Quality:**

3

**Strengths And Weaknesses:**

**Strengths**

S1: Novel problem formulation

The paper addresses an important gap in synthetic EHR generation by focusing on raw multi-relational table rather than preprocessed tables, which increases the accessibility of synthetic generation for general users.

S2: Technical innovation

The combination of text-based representation, compression via Residual Quantization, and temporal modeling creates an elegant solution to a complex problem. The authors demonstrate why RQ is superior to VQ for their task through careful ablation studies.

S3: Strong empirical results

RawMed consistently outperforms baselines across metrics


**Weaknesses**

W1: Lack of conditional generation

The method generates data unconditionally, whereas many practical applications would benefit from the ability to generate synthetic EHRs conditioned on specific patient characteristics or medical conditions. It would be beneficial to conduct small conditional generation tasks

W2: If post-processing makes the problem complicated?

In the paper, it says "These text-based events and timestamps are then converted to relational tables." Is this step heuristic? Could authors explain and investigate the robustness of algorithm 1 in appendix.

---

> ### Author Rebuttal · Authors · 2025-07-31
>
> We sincerely appreciate your insightful feedback and constructive comments, which are invaluable in improving our work.
>
> **Response to Weakness1 & Question2:**
> You've rightly pointed out the importance of conditional generation for practical applications. We agree entirely. Our initial focus was to establish the model's fundamental generation capabilities. However, we've conducted preliminary experiments on conditional generation incorporating static features during this rebuttal period.
>
> Specifically, we integrated age, gender, and admission diagnosis into the Temporal Modeling (Section 3.3). For training, we prepended these static features to the sequences, and the model learned the combined sequence auto-regressively. During generation, the model uses real static features as a condition to produce the complete sequence. To further explore conditional generation, we also experimented with providing initial medical context as a condition. This involved providing the model with real static features and approximately one-fourth of the initial real-data events from the sequence, with the model then generating the remaining three-fourths of the sequence.
>
> To evaluate if the generated sequences effectively captured the relationship with static features, we used three classifiers to predict each static feature (age, gender, admission diagnosis) from the generated sequences, excluding the static feature portion itself. Age was binned for this classification. Model performance was evaluated using micro-AUROC for multiclass classifications (age and admission diagnosis) and AUROC for binary classification (gender).
>
> | Model                          | Age  | Gender | Admission Diagnosis |
> |--------------------------------|------|--------|---------------------|
> | Real                           | 0.72 | 0.67   | 0.89                |
> | RawMed (static only)           | 0.65 | 0.52   | 0.86                |
> | RawMed (static+1/4 initial events) | 0.69 | 0.59   | 0.88                |
>
> These results show comparable performance for age and admission diagnoses between real and synthetic data, especially when leveraging the initial medical context. The increased AUROC values when conditioning on a portion of initial events (e.g., 0.69, 0.59, 0.88 for age, gender, and admission diagnosis, respectively, compared to 0.65, 0.52, 0.86 with static features alone) experimentally confirm our framework's flexibility in effectively supporting conditional generation using a richer set of features, leading to improved synthetic data quality. While we observed a slight drop in gender prediction compared to real data, we believe there's room for improvement with further methodological enhancements and hyperparameter tuning, given the short experimental timeframe. Due to these constraints, experiments were limited to the eICU dataset; we will update with MIMIC-IV results (for age, gender, and admission type) during the discussion period.
>
> Regarding the extension to a hybrid generation mode with user preferences, this is indeed a very interesting direction. It would be feasible to fine-tune RawMed using techniques such as Direct Preference Optimization (DPO) [1] with a separate dataset that contains both "preferred synthetic data" and "un-preferred synthetic data." This aligns with our future research interests in developing more controllable and customizable synthetic data generation.
>
>
> ---
>
>
> **Response to Weakness2:**
> This process is not heuristic but rather a rule-based approach that adheres to the minimum requirements of time-series relational tables, including consistent table naming, column naming, preservation of column-value pairs, and adherence to the temporal ordering of events. By predefining the table names, column names, and data types (e.g., numeric or categorical) for EHR datasets, Algorithm1 can be applied to diverse datasets without additional tuning or modifications. Indeed, the same code was successfully applied to the MIMIC-IV, eICU, and MIMIC-IV-ED datasets.
>
> ---
>
> **Response to Question1:**
> The errors observed during the conversion of text to tabular format, as detailed in Algorithm 1, are primarily syntactic inconsistencies. These include incorrect table names, non-existent column-value pairs, mistyped column names, and mismatched data types. For example, a column that should be named "value" might appear as "value value", or "originalrate" might be truncated to "original." Additionally, a numeric column might contain non-numeric values like "1..23" or "8. 5. 6." instead of valid numerical entries. We posit that these errors originate from the auto-regressive modeling process within the compressed text space. These syntactic inconsistencies are not unique to RawMed. Other text-based autoregressive models for tabular data encounter similar errors. For example, the GReaT paper also discards syntactically invalid samples, highlighting a shared challenge in this domain.
>
>
> ---
>
>
> **Response to Question3:**
> When compressing raw EHR data with Residual Quantization (RQ), our method provides acceptable overall reconstruction accuracy on the test set. We achieved 99.8% token-level accuracy on eICU and 98.2% on MIMIC-IV, indicating that the data's general structure and core information are well-preserved across both datasets.
> Specifically, for eICU, table name reconstruction accuracy was 99.9%, column names 99.9%, categorical column values 99.8%, and numerical column values an estimated 99.4%. For MIMIC-IV, table name reconstruction accuracy was 99.9%, column names 99.9%, categorical column values 99.8%, and numerical column values an estimated 91.2%. While the reconstructed data might not perfectly match the original, the preserved information typically remains adequate for most data utilization purposes.
>
>
> ---
>
> **Reference**
>
> [1] Direct Preference Optimization: Your Language Model is Secretly a Reward Model (Rafailov et al.)

---

> > ### Comment · Reviewer_tPMX · 2025-08-06
> >
> > Thank authors for the detailed rebuttal and effort. I appreciate your clarifications which addressed some concerns of mine. I will keep my score.

---

> > > ### Author Response · Authors · 2025-08-06
> > > **Response to Reviewer Comments**
> > >
> > > Thank you for your response. We are glad that our rebuttal addressed some of your concerns. We appreciate you taking the time to review our response.
> > >
> > > If there are any remaining concerns that our rebuttal did not fully address, please let us know. We are ready to do our best to provide further clarification.

---

> > > > ### Author Response · Authors · 2025-08-08
> > > > **Update on Conditional Generation Experiments**
> > > >
> > > > During the rebuttal, we presented preliminary results for conditional generation on the eICU dataset, incorporating static features like age, gender, and admission diagnosis. As promised, we have now completed the experiments on the MIMIC-IV dataset and would like to share the updated results. Model performance was evaluated using micro-AUROC for multiclass classifications (age and admission type) and AUROC for binary classification (gender).
> > > >
> > > > | Model                            | Age  | Gender | Admission Type |
> > > > |----------------------------------|------|--------|---------------------|
> > > > | Real                             | 0.80 | 1.00   | 0.92                |
> > > > | RawMed (static only)             | 0.75 | 1.00   | 0.89                |
> > > > | RawMed (static+1/4 initial events) | 0.77 | 1.00   | 0.89                |
> > > >
> > > > Similar to our observations with the eICU dataset, the synthetic data shows comparable performance to the real data, demonstrating that our model effectively captures the relationship between static features and sequential data. We also observe that conditioning on initial events leads to improved performance, with a higher AUROC for age prediction compared to using only static features.
> > > > Specifically for the gender prediction task on MIMIC-IV, we observed an AUROC score of 1.00. We have identified that this is because certain lab values in the *labevents* table (*e.g.*, creatinine, hemoglobin) have gender-specific reference ranges in the *ref_range_lower* and *ref_range_upper* columns. These values act as direct indicators of a patient's gender, which is reflected in the prediction scores for both the real and synthetic data.
> > > > The results on the MIMIC-IV dataset align with our previous eICU observations, and further validate our model's flexibility and potential for conditional generation.
> > > >
> > > > We hope these updated results are helpful. Thank you again for your valuable time and feedback throughout this review process.

---

### Official Review · Reviewer_4p1X · 2025-07-03

**Clarity:** 3
**Significance:** 3
**Originality:** 2
**Rating:** 4
**Confidence:** 3

**Summary:**

This paper presents RawMed, designed specifically for multi-table time-series Electronic Health Records (EHRs). Unlike previous methods that rely heavily on feature selection and complex preprocessing, this work synthesizes raw, minimally processed EHR data, including all original table columns and values. It uses a text-based representation of EHR events and compresses these representations into a latent space via Residual Quantization (RQ), enabling efficient autoregressive generation.

**Questions:**

See above,

**Ethical Concerns:**

["NO or VERY MINOR ethics concerns only"]

**Limitations:**

Yes.

**Quality:**

3

**Strengths And Weaknesses:**

Strength:
- This study addresses an important, under-explored challenge: generating multi-table, time-series EHR data with minimal preprocessing, preserving original values and database schema integrity.
- The paper introduces a well-rounded evaluation framework that thoroughly assesses synthetic data across multiple critical dimensions

Weakness:
- The evaluation is limited to only two datasets (MIMIC-IV and eICU). The generalizability and robustness of RawMed would benefit from additional evaluation on more diverse datasets.
- The compression and subsequent decompression of event embeddings may introduce distortions or inaccuracies. I am curious whether the compressed time series representation can be decoded to the original information. For example, if users ask about the height of a patient, can the model answer it?

---

> ### Author Rebuttal · Authors · 2025-07-31
>
> **Response to Weakness1:**
> We appreciate the reviewer’s concern regarding the evaluation of RawMed on only two datasets (MIMIC-IV and eICU) and the suggestion to include more diverse datasets to assess generalizability. However, as noted in Section 4.1 (Setup: Dataset, lines 201–203 in the manuscript), RawMed was also evaluated on the MIMIC-IV-ED dataset, which represents emergency department settings and provides a distinct clinical context compared to the ICU-based MIMIC-IV and eICU datasets. The results, reported in Appendix F.3 and Table 16, show that RawMed’s performance closely matches real data utility and outperforms baseline methods, indicating its potential for generalization to other datasets despite different clinical environments.
>
> **Response to Weakness2:**
> Thank you for your insightful comments. To empirically quantify the deviation from original information that might occur during the compression of event embeddings, we measure the token-level reconstruction accuracy on the test set, reaching 98.2% for MIMIC-IV and 99.8% for eICU. This confirms our compression introduces only negligible distortion, barely affecting the subsequent autoregressive decoding stage, while considering the benefits of compression for text-based embeddings.
>
> As detailed in 3.2 and 3.3, our method compresses individual event representations, not the entire time-series representation. Therefore, our model does not involve decoding a compressed time-series back to its original form. Instead, we focus on reconstructing each individual event representation.
>
> Regarding your last question about querying patient height, we would greatly appreciate it if you could elaborate on your question. Our model is designed for synthetic EHR generation, not for question-answering tasks, and is not required to address such direct inquiries.

---

> > ### Comment · Reviewer_4p1X · 2025-08-08
> >
> > Thanks authors for the clarification, my concerns about the evaluation and compression parts are addressed. I will keep my score.

---

> > > ### Author Response · Authors · 2025-08-09
> > > **Response to Reviewer Comments**
> > >
> > > We're glad we could address your concerns. Thank you for taking the time to review our rebuttal.

---

> ### Author Response · Authors · 2025-08-07
> **Discussion on Rebuttal**
>
> Thank you for your valuable review. We wanted to respectfully follow up on our rebuttal. As the discussion period is nearing its end, we would appreciate it if you could let us know if our responses have addressed your concerns. If there are any points that still need clarification, we would be glad to provide further details.

---

### Official Review · Reviewer_Wqvt · 2025-07-05

**Clarity:** 2
**Significance:** 3
**Originality:** 3
**Rating:** 4
**Confidence:** 3

**Summary:**

This paper proposed a framework, named RawMed to generate raw multi-table time-series EHRs and also introduce a evaluation framework for the synthetic raw EHRs which covers all features of the raw EHRS. Experiments on two public datasets validate the proposed method.

**Questions:**

refer to strength and weakness

**Ethical Concerns:**

["NO or VERY MINOR ethics concerns only"]

**Final Justification:**

The authors addressed most of my concerns, considering the paper's novelty and promising results, I keep my score as 4.

**Limitations:**

yes

**Quality:**

3

**Strengths And Weaknesses:**

Strength:
1. The major strength is that the paper is the first work designed to synthesize raw, multi-table time-series EHR, and requiring minimal preprocessing in the method (e.g., compression module).
2. The corresponding eval framework is novel covering the complex temporal and inter-table dynamics.
3. The experimental validation is thorough and convincing.

Weakness:
1. As discussed in the paper, the paper only targets on limited scope of the tables used in the experiments, not sure if it will be able to scale up.
2. The handling of static patient attributes is a limitation, which doesnot explicitly model static features like gender or age.
3. Post-processing steps are still needed in the work to ensure the structural and temporal integrity of the generated data.

---

> ### Author Rebuttal · Authors · 2025-07-31
>
> **Response to Weakness1&2:**
>
> We sincerely appreciate your insightful feedback particularly in highlighting the current limitations regarding the scope of tables and the handling of static patient attributes. As we explicitly discussed in the Section 6 of our paper, these are the current limitations of RawMed. In this initial stage of our research, our primary focus was to establish the model's fundamental generation capabilities. We acknowledge these as key areas for future work, aiming to scale up to dozens of tables, including those with static features.
>
> During the rebuttal period, we conducted preliminary experiments on conditional generation incorporating static features. Specifically, we integrated age, gender, and admission diagnosis into the Temporal Modeling (Section 3.3).
>
> For training, we prepended these static features to the sequences, and the model learned the combined sequence auto-regressively. During generation, the model uses real static features as a condition to produce the complete sequence.
>
> To further explore conditional generation, we also experimented with providing initial medical context as a condition. This involved providing the model with real static features and approximately one-fourth of the initial real-data events from the sequence, with the model then generating the remaining three-fourths of the sequence.
>
> To evaluate if the generated sequences effectively captured the relationship with static features, we used three classifiers to predict each static feature (age, gender, admission diagnosis) from the generated sequences, excluding the static feature portion itself. Age was binned for this classification. Model performance was evaluated using micro-AUROC for multiclass classifications (age and admission diagnosis) and AUROC for binary classification (gender).
>
> | Model                          | Age  | Gender | Admission Diagnosis |
> |--------------------------------|------|--------|---------------------|
> | Real                           | 0.72 | 0.67   | 0.89                |
> | RawMed (static only)           | 0.65 | 0.52   | 0.86                |
> | RawMed (static+1/4 initial events) | 0.69 | 0.59   | 0.88                |
>
> These results show comparable performance for age and admission diagnoses between real and synthetic data, especially when leveraging the initial medical context. The increased AUROC values when conditioning on a portion of initial events (e.g., 0.69, 0.59, 0.88 for age, gender, and admission diagnosis, respectively, compared to 0.65, 0.52, 0.86 with static features alone) experimentally confirm our framework's flexibility in effectively supporting conditional generation using a richer set of features, leading to improved synthetic data quality. While we observed a slight drop in gender prediction compared to real data, we believe there's room for improvement with further methodological enhancements and hyperparameter tuning, given the short experimental timeframe. Due to these constraints, experiments were limited to the eICU dataset; we will update with MIMIC-IV results (for age, gender, and admission type) during the discussion period.
>
>
> ---
>
>
> **Response to Weakness3:**
> The need for post-processing steps to ensure the structural and temporal integrity of generated data is a valid and widely accepted practice for quality assurance across various domains. These steps are (1) inevitable, as an autoregressive generation—predicting tokens sequentially—inherently accumulates errors over time, making it challenging to produce a 100% flawless sample in a single pass. For example, the text-based tabular synthesis model GReaT [1], which relies on autoregressive generation, discards samples that violate required formats. Moreover, they are (2) practical, as even renowned generative models in the image domain, such as VQGAN [2] and VQ‑VAE2 [3],use classifier-based rejection sampling to filter out low-quality samples and improve overall sample quality.
>
> [1] Language Models are Realistic Tabular Data Generators (Borisov et al., ICLR 2023)
> [2] Taming Transformers for High-Resolution Image Synthesis (Esser et al., CVPR 2021)
> [3] Generating Diverse High-Fidelity Images with VQ-VAE-2 (Razavi et al., NeurIPS 2019)

---

> > ### Comment · Reviewer_Wqvt · 2025-08-05
> >
> > Thanks for the rebuttal.
> >
> > 1. The experiments and analysis on the static feature makes sense to me. It's better to include it to the main manuscript.
> > 2. As for post-processing, I insist on my opinion.

---

> > > ### Author Response · Authors · 2025-08-05
> > > **Response to Reviewer Comments**
> > >
> > > Thank you for your valuable feedback. We are grateful for your positive feedback of our experiments and analysis on static features, and we will incorporate these findings into the paper as recommended to strengthen the paper.
> > >
> > > We understand and respect your concerns regarding post-processing. While the ideal is to generate flawless data in a single pass, the inherent limitations of autoregressive models often necessitate post-processing to ensure data quality, a practice widely adopted in the field.
> > >
> > > To better address your perspective, could you please clarify specific aspects of our post-processing approach that concern you? This will enable us to provide a more detailed response or discuss potential improvements.

---

> > > > ### Author Response · Authors · 2025-08-08
> > > > **Update on Conditional Generation Experiments**
> > > >
> > > > During the rebuttal, we presented preliminary results for conditional generation on the eICU dataset, incorporating static features like age, gender, and admission diagnosis. As promised, we have now completed the experiments on the MIMIC-IV dataset and would like to share the updated results. Model performance was evaluated using micro-AUROC for multiclass classifications (age and admission type) and AUROC for binary classification (gender).
> > > >
> > > > | Model                            | Age  | Gender | Admission Type |
> > > > |----------------------------------|------|--------|---------------------|
> > > > | Real                             | 0.80 | 1.00   | 0.92                |
> > > > | RawMed (static only)             | 0.75 | 1.00   | 0.89                |
> > > > | RawMed (static+1/4 initial events) | 0.77 | 1.00   | 0.89                |
> > > >
> > > > Similar to our observations with the eICU dataset, the synthetic data shows comparable performance to the real data, demonstrating that our model effectively captures the relationship between static features and sequential data. We also observe that conditioning on initial events leads to improved performance, with a higher AUROC for age prediction compared to using only static features.
> > > > Specifically for the gender prediction task on MIMIC-IV, we observed an AUROC score of 1.00. We have identified that this is because certain lab values in the *labevents* table (*e.g.*, creatinine, hemoglobin) have gender-specific reference ranges in the *ref_range_lower* and *ref_range_upper* columns. These values act as direct indicators of a patient's gender, which is reflected in the prediction scores for both the real and synthetic data.
> > > > The results on the MIMIC-IV dataset align with our previous eICU observations, and further validate our model's flexibility and potential for conditional generation.
> > > >
> > > > As the discussion period is nearing its end, we would be grateful if you could let us know if there are any remaining concerns or points that require further clarification. We are ready to provide a prompt response.

---

> > > > ### Comment · Reviewer_Wqvt · 2025-08-09
> > > >
> > > > I acknowledge that post-processing is necessary sometimes, for example, for integrity. However, if it is to make the generation quality higher, stronger one-pass model is a better choice.

---

> ### Author Response · Authors · 2025-08-09
> **Response to Reviewer Comments**
>
> We appreciate your valuable feedback. We'd like to clarify our post-processing approach, as there may have been a misunderstanding. Our method does not use rejection sampling to select superior samples from a pool of generated candidates. Instead, our post-processing is focused on guaranteeing the required data integrity for real-world applications.
>
> To be specific, our post-processing serves two distinct purposes. First, it is a necessary step to convert the model's generated text into a tabular format. Second, it ensures the structural and temporal integrity of the data. For example, any sample that violates a specified min-max range for a column is discarded.
> We hope this clarification addresses your concerns and provides a clearer understanding of our methodology.

---

> > ### Author Response · Authors · 2025-08-09
> > **Response to Reviewer Comments**
> >
> > Thank you for your time and continued focus on our work. We'd like to provide further clarification, as it seems our previous response didn't fully address your concerns about our post-processing.
> >
> > To be clear, of the two post-processing purposes you mentioned—'for data integrity' and 'to improve generation quality'—we want to emphasize that our method performs the former.
> >
> > Our post-processing is an essential step to ensure the integrity of the generated data, serving two main purposes: first, to convert the generated text into a proper tabular format; and second, to ensure the structural and temporal integrity of the data. For example, any sample that violates a specified min-max range for a column is discarded.
> >
> > We hope this clarification addresses your concerns. If you have any further questions, please do not hesitate to let us know. We'll be available until the deadline to provide any further detail needed. Thank you again for your valuable time and consideration.

---

### Note · Authors · 2025-08-12

Thank you to all reviewers for their dedicated engagement and constructive feedback throughout the rebuttal and discussion period. We would like to summarize the main points of discussion.

We addressed Reviewer 4p1X's concerns about generalizability by clarifying that our evaluation already included a distinct dataset (MIMIC-IV-ED). We also resolved questions about information loss by presenting reconstruction accuracy metrics, which demonstrated minimal loss during our compression process.

Both Reviewer tPMX and Reviewer Wqvt pointed out the lack of conditional generation as a weakness. We are grateful for their positive feedback on our preliminary experiments, which showed promising results when incorporating static features. Additionally, we clarified our post-processing steps to address Reviewer tPMX's concerns about their robustness.

A discussion with Reviewer Wqvt centered on the necessity of post-processing. While we agree with the ideal of generating flawless data that requires no post-processing, we maintain that post-processing is a pragmatic and essential step for autoregressive models to ensure data integrity. As we clarified, our post-processing is for converting text to a proper tabular format and enforcing structural and temporal integrity, not for general quality improvement. We believe this clarification aligns our perspective with the reviewer's final comments, thus addressing the initial disagreement.

Finally, Reviewer Uzbn's extensive feedback was instrumental in refining our claims. The discussion on "minimal preprocessing" was particularly valuable, as we clarified that the term was not meant to suggest little or no preprocessing. Instead, the term emphasizes our method’s avoidance of lossy steps—such as feature selection and binning—and instead employs universal, near-lossless transformations that preserve the raw data’s integrity. In response to reviewer's questions, we also conducted an analysis on post-processing and top-k sampling, which will significantly improve the paper's rigor. The additional clarifications provided in the final response have settled the remaining concerns.

We are pleased that the discussions with Reviewers 4p1X, tPMX, and Uzbn successfully addressed their concerns, as reflected in their final comments. While there was an initial difference of opinion with Reviewer Wqvt, we believe our final response has aligned with their perspective.

---

### Decision · Program_Chairs · 2025-09-17

**Decision:**

Accept (poster)

**Comment:**

This paper introduces RawMed, a framework for generating raw, multi-table, time-series EHRs with minimal preprocessing, alongside a comprehensive evaluation suite that spans fidelity, temporal dynamics, inter-table dependencies, downstream clinical utility, and privacy. The reviewers consistently highlight the novelty of the problem formulation and the breadth of the evaluation framework as the main strengths. The empirical validation on MIMIC-IV and eICU demonstrates convincing gains over adapted baselines, particularly in modeling temporal fidelity and preserving raw data structure.

Overall, while the technical novelty lies more in the integration of known components than in the invention of new generative methods, the framing of the problem, the evaluation toolkit, and the demonstrated empirical results represent a meaningful and timely contribution to synthetic healthcare data research. On balance, the strengths outweigh the weaknesses, and I recommend acceptance.